# Postnatal development of retrosplenial projections to the parahippocampal region of the rat

**Jørgen Sugar, Menno P Witter\***

Kavli Institute for Systems Neuroscience and Centre for Neural Computation, Norwegian University for Science and Technology, Trondheim, Norway

**Abstract** The rat parahippocampal region (PHR) and retrosplenial cortex (RSC) are cortical areas important for spatial cognition. In PHR, head-direction cells are present before eye-opening, earliest detected in postnatal day (P)11 animals. Border cells have been recorded around eye-opening (P16), while grid cells do not obtain adult-like features until the fourth postnatal week. In view of these developmental time-lines, we aimed to explore when afferents originating in RSC arrive in PHR. To this end, we injected rats aged P0-P28 with anterograde tracers into RSC. First, we characterized the organization of RSC-PHR projections in postnatal rats and compared these results with data obtained in the adult. Second, we described the morphological development of axonal plexus in PHR. We conclude that the first arriving RSC-axons in PHR, present from P1 onwards, already show a topographical organization similar to that seen in adults, although the labeled plexus does not obtain adult-like densities until P12.

**\*For correspondence:** menno. witter@ntnu.no

**Competing interests:** MPW: Member of the board of the Kavli Centre, and of the scientific advisory board of the Center for Behavioral Brain Sciences, Otto von Guericke University, Magdeburg, FDR. The other author declares that no competing interests exist.

## Introduction

The parahippocampal region (PHR) is important for learning and memory. It consists of two functionally different networks, one of which, involved in spatial functions, comprises the presubiculum (PrS), parasubiculum (PaS), medial entorhinal cortex (MEC) and postrhinal cortex (POR). The other important for object representation and processing of contextual information, comprising lateral entorhinal cortex (LEC) and perirhinal cortex (PER; *Eichenbaum et al., 2007*; *Canto et al., 2008*; *Ranganath and Ritchey, 2012*; *Knierim et al., 2014*; *Witter et al., 2014*). PrS, PaS, and MEC harbor cells representing an animal's position (grid cells), direction (head direction cells), borders in the environment (border cells) and speed of the animal (speed cells; *Fyhn et al., 2004*; *Solstad et al., 2008*; *Boccara et al., 2010*; *Kropff et al., 2015*). Even though a complete mechanistic understanding on how these spatial codes emerge is still lacking, it is believed that both intrinsic connectivity and extrinsic afferents are necessary to produce the receptive fields observed (*Brandon et al., 2011*; *Koenig et al., 2011*; *Bonnevie et al., 2013*; *Couey et al., 2013*; *Newman et al., 2014*). One approach applied to identify the critical elements underlying the functioning of these cell types, has been to study the developmental aspects of PHR networks. The different spatially modulated neuron types in PHR emerge at different periods during development. Border cells and head-direction cells can both be observed at the end of the second- and beginning of the third postnatal week (*Bjerknes et al., 2014*; *Bjerknes et al., 2015*; *Tan et al., 2015*). In contrast, adult-like grid cells in layer II of MEC first appear during the fourth postnatal week (*Langston et al., 2010*; *Wills et al., 2010*). The early presence of head-direction cells is apparently paralleled by a similarly early developed shared connectivity (*Bjerknes et al., 2015*), while the late development of grid cells is paralleled by a corresponding late development of the relevant intrinsic connectivity in MEC (*Langston et al., 2010*; *Couey et al., 2013*). Regarding main inputs, sparse connections between

**eLife digest** Our ability to navigate critically depends on part of the brain called the parahippocampal region. Within this region, there are several different types of brain cells (or neurons) whose activity "codes" different aspects of navigation, such as position, direction and speed.

To understand how parahippocampal neurons are able to form these activity patterns, we need to understand how they develop connections with neurons from other brain regions that are important for navigation, such as the retrosplenial cortex. If inputs from retrosplenial neurons are important for generating the activity patterns observed in the parahippocampal region, the connections between the two groups of neurons should be fully mature before the activity patterns emerge. In rats, this should occur around 11–16 days after birth.

Sugar and Witter have now assessed how the retrosplenial inputs are organized in the parahippocampal region of rats. This revealed that, when the rats are born, there are very few retrosplenial inputs present in the parahippocampal region. However, the few inputs that are present are organized similarly to how they eventually will be organized in adults. After birth, the number of inputs gradually increases until the rats are approximately 12 days old, at which point the pattern of connections is indistinguishable from what we observe in adults. Thus it appears that retrosplenial inputs are fully mature before activity patterns emerge in the parahippocampal region.

In the future, Sugar and Witter would like to investigate how inputs to the parahippocampal region are able to organize themselves during early development. The importance of retrosplenial inputs could also be investigated by manipulating them during development and adulthood.

the hippocampal formation (HF) and PHR are present from birth, reaching adult-like morphological features during the second postnatal week (*Deng et al., 2007*; *O'Reilly et al., 2013*; *O'Reilly et al., 2015*). A similar developmental timeline has also been reported for functional connections from PrS/PaS to MEC (*Canto et al., 2011*). However, the timeline of development of cortical afferents to HF and PHR is still unknown.

One of the most prominent cortical inputs to PHR originates in the retrosplenial cortex (RSC) and spatially modulated cells have been found also in the latter cortical domain (*Cho and Sharp, 2001*; *Sugar et al., 2011*; *Alexander and Nitz, 2015*). Lesions of RSC result in impairments in navigational tasks (*Vann et al., 2009*). In addition, RSC in rodents is necessary for fear conditioning, both when context or complex multimodal stimuli are used as conditional stimuli (*Keene and Bucci, 2008a*; *2008b*; *Corcoran et al., 2011*; *Cowansage et al., 2014*; *Robinson et al., 2014*) and in rabbits RSC neurons are responsive to auditory cues when used as a CS in a memory task (*Gabriel et al., 1991*), suggesting that RSC has a general role in memory processes. The effect of lesioning RSC on navigation and memory performance is surprisingly similar to that seen after lesions that inflict the HF-PHR. It has thus been postulated that interactions between RSC and HF-PHR are crucial for spatial processing. In the adult, RSC projections target preferentially POR, PrS, PaS, and MEC (*Jones and Witter, 2007*; *Sugar et al., 2011*; *Kononenko and Witter, 2012*; *Czajkowski et al., 2013*). In this paper, we aimed to ascertain the relevance of the RSC-PHR projection for the development of the functionally different neuron types in PHR. We hypothesized that if inputs from RSC are important for the development of head-direction- and/or border cells, these inputs should be present before eye-opening. Alternatively, if inputs from RSC are only important for the formation of stable grid cells, these inputs might develop after eye-opening, likely reaching adult-like morphology during the third and fourth week. We injected RSC of rats at different postnatal ages, ranging from postnatal day 0 to 28 (P0-P28), with anterograde tracers. Using retrograde tracing, we identified RSC neurons originating the developing PHR projections. The anterograde experimental material was used to analyze the organization of RSC-PHR projections in postnatal rats and to compare these results with data previously obtained in the adult. We further analyzed the development of axon morphology and densities of axonal plexus in PHR.

## Results

### Nomenclature

Several nomenclatures have been used to describe subdivisions of RSC. These nomenclatures mainly follow the same cytoarchitectonic- and histochemical criteria and are therefore directly comparable. For a summary and direct comparison of the different nomenclatures, we refer to a recent review on the RSC-HF-PHR connectivity (*Sugar et al., 2011*). In the current manuscript, we chose to use the nomenclature of *Vogt (2004)*. We defined the border between area (A)29 and A30 as the area where layer II changes from being very condensed in A29, to less condensed in A30. The rostral border of RSC, towards the anterior cingulate cortex (ACC) was defined as the area where layer II/III widens and where layer IV shifts from being clearly demarcated in RSC towards being more diffusely organized in ACC. The ventral border of RSC with PHR, more specifically with PrS, was defined by the appearance of a cell free lamina dissecans in PrS, not present in RSC. In adult rats, A29 can be further subdivided into three different cytoarchitectonic subdivisions; A29a, b and c, which are involved in different cognitive functions (*van Groen et al., 2004*). However, in the immature cortex, these cytoarchitectonic areas are not apparent. We therefore chose to define a continuous measure of the dorsoventral positions within RSC (see methods for details). This measure is indirectly related to the classical cytoarchitectonic subdivision since the classical borders of A29a, b and c follow approximately our continuous definition of the dorsoventral axis of RSC.

### Injection sites

To investigate the development of RSC projection patterns in PHR we aimed to inject anterograde tracers in different locations within RSC of differently aged pups. Of the 82 animals used in this study, 20 animals either did not survive surgery or no injection sites were observed in RSC. In the remaining 62 animals we obtained 113 injections in RSC. Eight of these injections only involved layers I and/or layers II-III and did not result in any labeled fibers in HF-PHR. These experiments were therefore excluded from further analyses. The remaining 105 injections all covered at least parts of layer V of RSC and involved different parts of RSC. We obtained one (n=21), two (n=31), three (n=6) or four (n=1) ipsilateral injections in RSC of each brain (*Figure 1*). In our analyses, we regarded each of these injections as independent experiments. Most of the experiments (n=93) were performed in animals aged P15 or younger since other comparable corticocortical projections are developed before eye-opening and the functional cell types are present in HF-PHR at this age (*Langston et al., 2010*; *O'Reilly et al., 2013*; *Bjerknes et al., 2014*; *O'Reilly et al., 2014*). However, we also obtained injections in older pups (n=12) with a maximum age of P28.

To compare the location of injections in brains of different ages, we age-normalized the position of the injections by the use of a 3D-atlas brain (see methods and *Figure 1A*, *Video 1* and *2*). This was achieved by identifying atlas-sections containing landmarks and cytoarchitectonic borders present in the histological section containing the center of each injection. In the atlas-section, we recorded the coordinate of the center of each injection. Since the caudal RSC cortex is curved both along the dorsoventral and rostrocaudal axis we divided the pial surface of RSC in the atlas brain into triangles and used these triangles to calculate normalized rostrocaudal and dorsoventral coordinates of the each injection (*Figure 1B and C*). For visualization, we plotted the normalized injection coordinates into a schematic representation of RSC (*Figure 1D*).

### General projection patterns

We analyzed the anterograde labeling resulting from all 105 injections. In some (n=44), we observed a few retrogradely labeled neurons within the dorsal half of subiculum (SUB). This potentially may lead to false positive labeling, which will be addressed in the detailed descriptions below. In accordance with previous studies in adults, we observed anterogradely labeled fibers in the striatum, anterior nuclei of the thalamus, anterior cingulate cortex, parietal cortex and visual cortices and in the brainstem (*van Groen and Wyss, 1990*; *1992*; *2003*; *Jones et al., 2005*). However, a detailed assessment of these projections is outside the scope of this paper and here we will detail the projections to PHR. We observed anterogradely labeled fibers in layers I, III and V-VI of PrS (in all of the experiments included in the analyses, n=105), layers V-VI of PaS (n=90), layers V-VI of MEC (n=90), layers V-VI of medial LEC (n=14) and in all layers of posterior POR (n=75). In some cases, we also

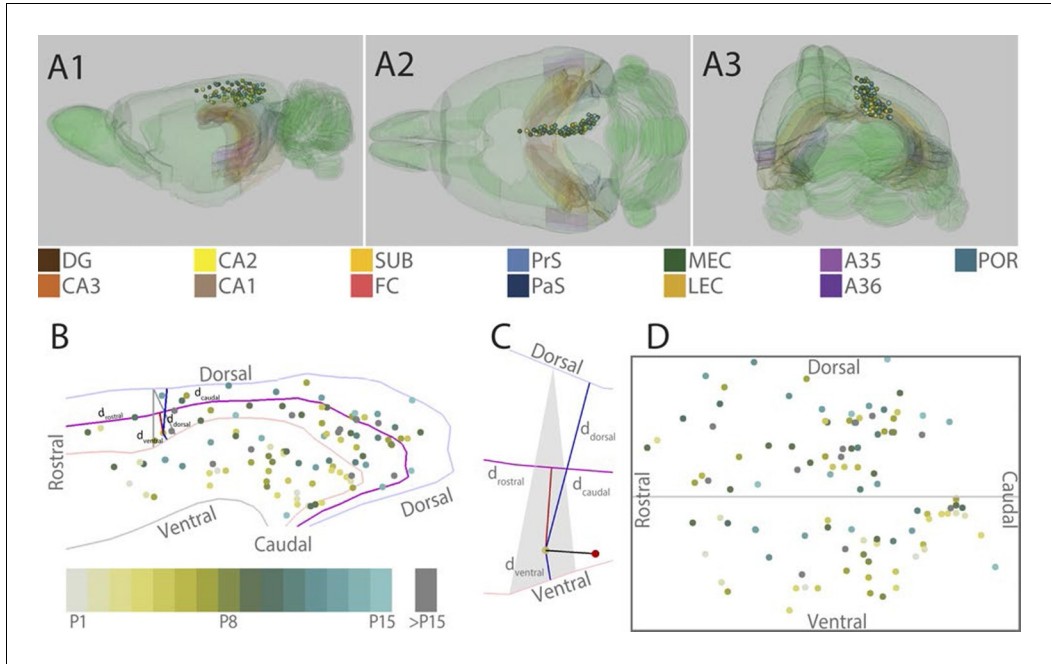

**Figure 1.** Location of the center of injections. (**A**) The location of the center of each injection was normalized to a standard 3D atlas of the rat brain (Waxholm space; **Papp et al., 2014**; See **video 1** and **2**). Lateral (A1), dorsal (A2) and para-caudal view (A3) of the 3D atlas brain with the center of each injection (colored spheres). The injections are color-coded according to age (see color code bottom left). Injections were performed in either the left or right hemisphere but to ease visualization, all injections were plotted in the right hemisphere. Dentate gyrus (DG), CA3-1, subiculum (SUB), fasciola cinereum (FC), pre- and parasubiculum (PrS and PaS), medial- and lateral entorhinal cortex (MEC and LEC), A35-36 and postrhinal cortex (POR) are color coded, while the rest of the brain is colored green. (**B**) Midsagital view of the center of the injections projected to the pial surface. The laminar position is disregarded to allow the injections to be plotted in 2D. Light blue, light red and grey line depict respectively the dorsal border of A30, the border between A29 and A30 and the ventral border of A29. Injections are color coded according to age. Triangle explained in C. (**C**) One example, as shown in B, of the algorithm used for calculating normalized 2D coordinates of the injections. The pial surface area of A29 and A30 was divided into triangles (grey area). The shortest vector (black line) between the injection (red sphere) and the cortical surface was calculated. Thereafter, we calculated the coordinate of the intersection of the vector and the plane within the triangle (yellow dot) which represented the 'transposed' location of the injection. The normalized dorsoventral coordinate of each injection was defined by calculating the shortest vector from the transposed injection to dorsal ($d_{dorsal}$) and ventral border of A29 or A30 ($d_{ventral}$). The normalized rostrocaudal coordinate was obtained by first calculating a line along the rostrocaudal extend of A29 and A30, positioned equally distant from the dorsal and ventral borders (magenta line). Next, we calculated the shortest vector between the injection and this line (red line) and found the intersection between the two. The rostrocaudal coordinate was obtained by calculating the cumulative distance from the cross section to the rostral ($d_{rostral}$) and caudal end of RSC ($d_{caudal}$). (**D**) Normalized flatmap of the injections. The 3D RSC is converted to a 2D normalized flatmap to obtain relative rostrocaudal and dorsoventral positions of the injections. The figure is oriented with rostral RSC (left), caudal RSC (right), dorsal RSC (top) ventral RSC (bottom) to each of the sides of the rectangle. Grey line depicts the border between A29 and A30 in RSC. In all figures, each injection is color coded according to the bottom left color scheme; light grey colored injections represent injections in pups aged P1, green colored injections represent injections in pups aged P8 while cyan colored injections represent injections in pups aged close to P15. Grey injections represent injections in pups older than P15.

observed single fibers in layers I and/or III of MEC (n=34). Additionally, we observed few fibers (if present typically one or two fibers in an experiment) in the dorsal half of SUB (n=41) and/or in the dorsal CA fields (n=16). In contrast to previously published data (**Burwell and Amaral, 1998b**), we did not observe any labeled fibers in PER in any of our experiments. This includes the oldest aged pups (P27-28) and the adult cases, suggesting that this lack of perirhinal projections is not a developmental feature. It is thus obvious that all experiments shared RSC-PHR projection patterns, but there

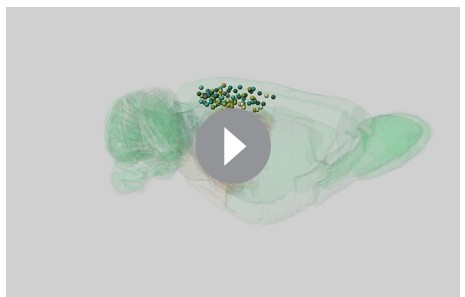

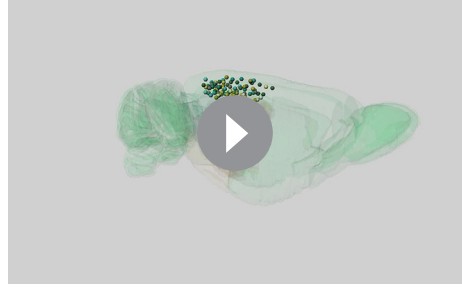

**Video 1.** Representation of all injection sites in RSC represented in a 3D rendering of the rat brain (Waxholm space; *Papp et al., 2014*; *2015*). The injection sites are color coded for age (for code see *Figure 1*)

**Video 2.** Representation of all injection sites in RSC represented in a 3D rendering of the rat brain (Waxholm space; *Papp et al., 2014*; *2015*). The injection sites are color coded for age (for code see *Figure 1*)

were also marked differences from case to case depending on the location of injection within RSC. This will be systematically described in the next sections.

## Injections in rostral A30

Injections in the rostral half of A30 (n=23) all resulted in comparable patterns of labeling in PHR. Labeled fibers were present in layers I, III and V-VI of the dorsal one-third of PrS. In layers I and III of PrS, labeled fibers were predominantly located in distal PrS and the densest plexus was generally observed in the dorsal extreme of PrS. Moderate numbers of labeled fibers were present in layers V-VI of dorsal PaS and layers V-VI of dorsomedial MEC. Among the injections in the rostral half of A30, the most caudal ones usually resulted in more widespread labeling in MEC. In none of the cases did we observe labeled fibers in the ventral two-thirds of PHR. We observed single labeled fibers in layer III of MEC in some of the experiments (n=3), in POR (n=16) and in HF (n=5).

In a representative animal (18433; P13), DA-A488 was injected in layers I-V in rostral A30 (*Figure 2A–D*, magenta). From the injection site, fibers continued caudally in layer VI of RSC and in the cingular bundle towards PHR. At the dorsal pole of PHR, fibers entered into the superficial layers and branched extensively in layer I and superficial layer III of intermediate proximodistal parts of the dorsal pole of PrS. At this dorsal level, labeled fibers were also present in layers V-VI of PrS. Single fibers continued into deep layers of dorsal PaS and deep layers of posterior POR and deep layers of dorsomedial MEC. No fibers were observed in more ventral levels of PHR, such that the ventral 75% of PHR did not show any labeled fibers.

Injections in rostral A30 thus resulted in labeling in the dorsal third of PHR, particularly in posterior POR, distal PrS, the complete transverse extent of PaS, and medial parts of MEC.

## Injections in caudal A30

We analyzed 32 injections in caudal A30. Overall, injections in caudal A30 resulted in comparable projection patterns. Labeled axons were observed in layers I, III and V-VI in distal parts of PrS, layers V-VI of PaS and in layers V-VI medially in MEC. Compared to the labelling observed after rostral injections, the labeling resulting from caudal injections extended more ventrally in PrS, PaS and MEC (compare magenta and green topography in *Figure 2C*). The total extend of all plexus covered approximately the dorsal half of PrS and PaS and the ventromedial part of MEC. The area receiving the densest projections was also shifted more ventrally. The maximum density of the plexus in PrS was usually located in the dorsoventral middle of its distal part. Only in one case, labeled fibers were present in the ventral one-third of PHR. At all dorsoventral levels, labeling covered the distal part of PrS, in layers I, III and V-VI, the proximodistal extent of PaS, and medial parts of MEC. After injections in caudal A30, we observed single labeled fibers in layer III of MEC in some experiments (n=12), in POR (n=29), and in HF (n=15).

In a representative animal (18453; P13), BDA was injected in layers I-V of intermediate-caudal A30 (*Figure 2A and D-E*, cyan). From the injection site labeled fibers continued caudally and ventrally in the cortex. Arriving in PHR, fibers continued ventrally in layer I and the lamina dissecans of PrS as

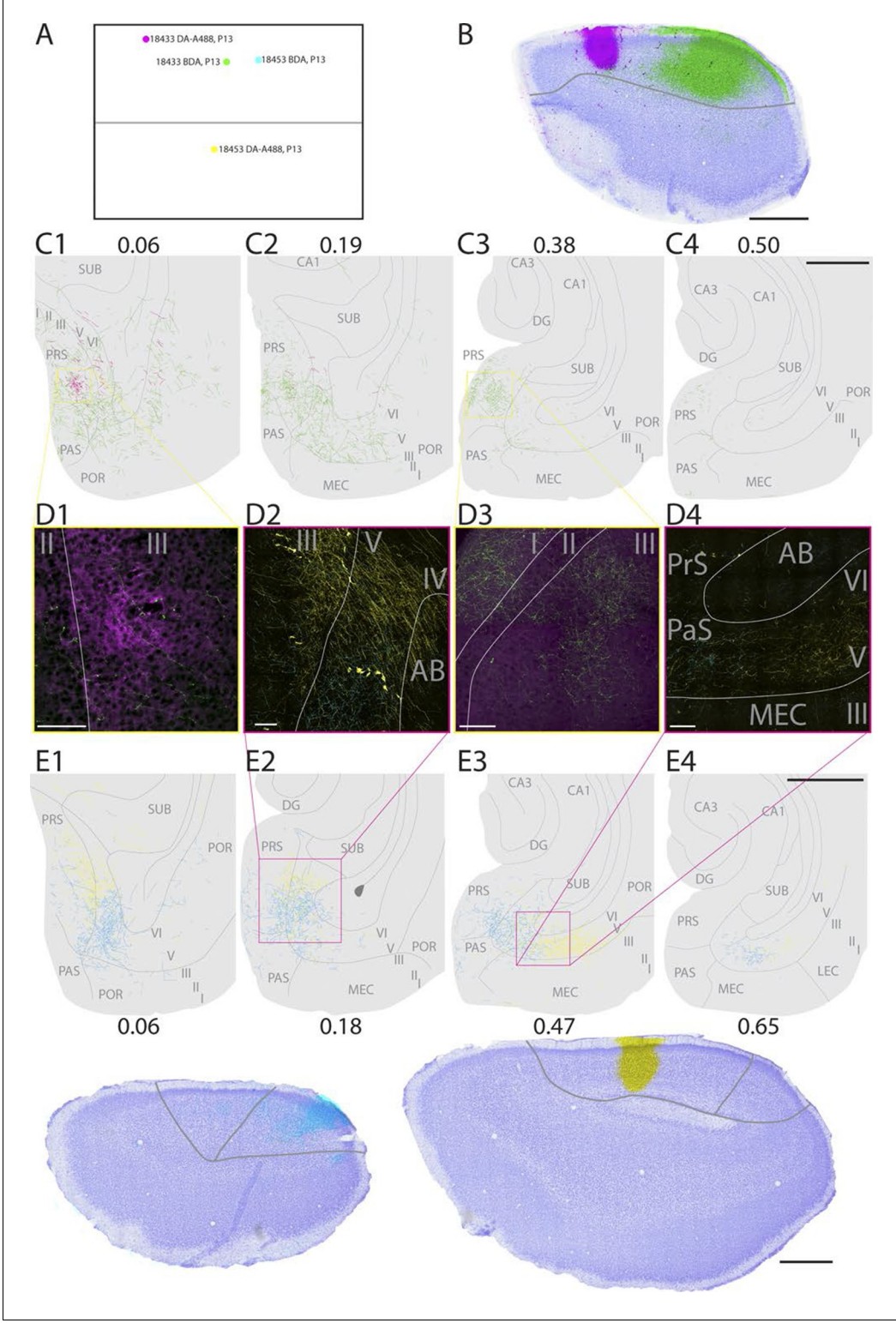

**Figure 2.** Representative examples of injections in RSC. (**A**) Normalized flatmap (see *Figure 1D*) of the locations of the injections in RSC, shown in **B** and **E**. Injections are located in the rostral A30 (magenta), intermediate rostral A30 (green), intermediate caudal A30 (cyan) and the intermediate rostral quarter of A29 (yellow). (**B**) Horizontally cut and Nissl stained section at the level of the injections overlaid with a neighboring fluorescent section containing the center of an injection in rostral A30 (magenta) and intermediate-rostral A30 (green) within the same animal. Grey line depicts delineation of A30. (**C**) The projections after the two injections shown in B were traced

*Figure 2 continued on next page*

*Figure 2 continued*

and represented in a dorsoventral series of drawings of horizontal sections through the PHR. After injections in rostral A30 (magenta) labeled fibers were mostly observed in the dorsal PrS layers I and III (C1, D1). After injections in intermediate-caudal A30 (green) the densest plexus was located more ventrally in PHR and in addition to labeled fibers in layers I, III and V-VI of PrS, labeled fibers also extended into layers V-VI of PaS, POR and MEC (C2-3 and D3). Numbers above sections indicate the dorsoventral position of the section relative to the total dorsoventral extent of PHR. The yellow boxes in C1 and C3 indicate the position of high power digital images obtained from the actual sections (D1 and D3). Grey lines depict borders between the HF-PHR subdivisions, the border between cortex and white matter and lamina dissecans. (D) High power images of plexus depicted in the sections shown in C and E. Roman numbers indicate cortical layers. Grey lines depict borders between layers. (D1) Labeled fibers in superficial layer III of PrS after injections in rostral A30 (magenta). Additionally a few fibers are seen originating in the intermediate-caudal quarter of RSC (green). (D2) Labeled fibers in proximal PrS deep layer III and layers V-VI after injection in intermediate-rostral A29 (yellow) and labeled fibers in distal PrS deep layer III and layers V-VI after injection in intermediate-caudal A30 (cyan). (D3) Labeled fibers in layers I and III after injection in intermediate-caudal A30 (green). No fibers originating in the rostral A30 were observed. (D4) After injection in intermediate-caudal A30 labeled fibers were observed in medial MEC (cyan), while after injection in intermediate-rostral A29 labeled fibers were observed in lateral MEC (yellow). (E) Top: the projections after two injections (bottom) were traced and represented in a dorsoventral series of drawings of horizontal sections through the PHR. After injections in intermediate-caudal A30 (cyan) labeled fibers were observed in distal PrS dorsally (E1-3). At more ventral levels fibers also extended into deep layers of PaS and medial MEC (E3-4). After injections in intermediate-rostral A29 (yellow) the densest plexus was located in proximal parts of PrS dorsally (E1-2). At more ventral levels the plexus in PrS layers I and III disappeared while in the deep layers the plexus shifted to lateral parts of EC at successively more ventral levels (E3-4). Numbers below sections indicate the dorsoventral position of the section relative to the total dorsoventral extent of PHR. The magenta boxes in E2 and E3 indicate the position of high power digital images obtained from the actual sections (D2 and D4). Grey lines depict borders between the HF-PHR subdivisions, the border between cortex and white matter and lamina dissecans. Bottom: Horizontally cut and Nissl stained sections at the level of the injection overlaid with neighboring fluorescent sections containing the center of an injection in intermediate-caudal A30 (cyan) and intermediate-rostral A29 (yellow) within the same animal. Gray line depicts delineation of A29 and A30. Scale bars equal 100 μm (high power images) and 1000 μm (low power images).

well as in deep white matter. At the dorsal pole of PrS, labeled fibers occasionally entered into layer III. A dense plexus was labeled in layers V-VI of the dorsal one-third of distal PrS, in layers V-VI of PaS and deep layers of medial POR. At more ventral levels, labeled fibers also extended into layers V-VI of MEC. In ventral parts of MEC, only a few labeled fibers were observed. No labeled fibers were observed in the most ventral one-third of PHR.

Injections in caudal A30 thus resulted in a labeling pattern in PHR comparable to that seen in case of rostral A30 injections, but extending to more ventral parts of PHR.

## Injections in rostral A29

Injections in the rostral half of A29 (n=15) resulted in a comparable pattern of labeling in PHR. Labeled fibers were present in the dorsal half of PrS layers I, III and V-VI, in layers V-VI of the dorsal half of PaS and in layers V-VI of MEC. One injection also resulted in labeled fibers in dorsal LEC. In PrS, the fibers tended to be located more proximally compared to the projections originating from A30 at the same rostrocaudal level (compare cyan and yellow fibers in *Figure 2E1–2*). Among the injections in rostral half of A29, the most caudal injections usually resulted in more extensive labeling of axons in MEC. In those cases, the plexus in MEC was located at intermediate mediolateral levels, more lateral compared to the MEC plexus seen after A30 injections (compare cyan and yellow fibers in *Figure 2E3–4*). In some cases, we also observed single labeled fibers in layer III of MEC (n=6), in POR (n=8) and in HF (n=9).

In a representative animal (18453; P13), DA-A488 was injected in layers I-VI of intermediate-rostral A29 (*Figure 2A and D–E*, yellow). From the injection site, labeled fibers ran caudally and ventrally in the cingular bundle and in layer VI of RSC. At the dorsal pole of PrS, labeled axons continued ventrally in layer I, lamina dissecans, and in the deep white matter. At this level, labeled fibers entered PrS and branched in layers I, III and V-VI of proximal PrS. Single fibers extended into deep layers of PaS and POR. At approximately the dorsoventral middle of PHR, the density of

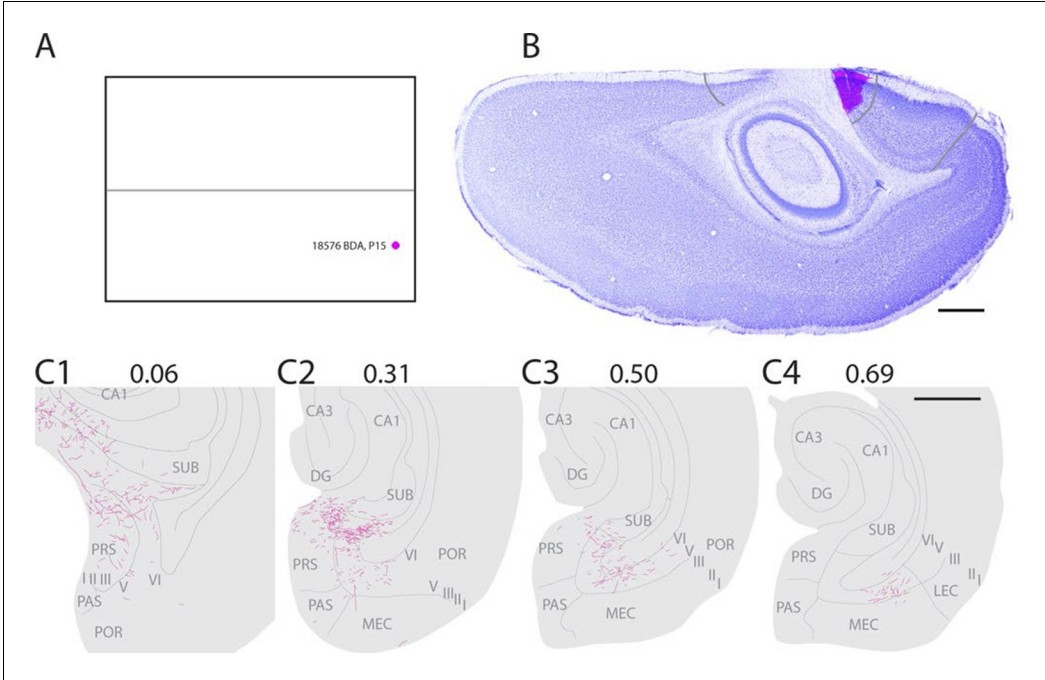

**Figure 3.** Representative example of an injection in caudal A29. (**A**) Normalized flatmap with location of an injection in caudal A29 (magenta). (**B**) Horizontally cut and Nissl stained section at the level of the injection overlaid with an adjacent section containing the center of the fluorescent tracer injection in caudal A29 (magenta). Grey lines depict delineation of A29 and A30. (**C**) The projections after the injection were traced and represented in a dorsoventral series of drawings of horizontal sections through the PHR. Labeled fibers are located in more proximal parts of layers I, III and V-VI of PrS compared to after projections in A30 (compare C1-3 and *Figure 2*). At successively more ventral levels, fibers also extended into increasingly more lateral parts of layers V-VI of MEC compared with injections in A30 (compare C2-4 and *Figure 2*). Grey lines depict borders between the HF-PHR subdivisions, the border between cortex and white matter and lamina dissecans. Scale bars equal 1000 µm.

labeled fibers in proximal PrS layers I and III gradually decreased, while in the deep layers, labeled fibers gradually shifted position at successively more ventral levels. More specifically, moving from dorsal to ventral levels of PHR, labeled fibers occupied proximal PrS at dorsal levels, and distal PrS, PaS, medial MEC, lateral MEC and finally LEC at successively more ventral levels.

It is thus apparent that injections in rostral A29 resulted in labeling mainly in the dorsal third of PHR, including posterior POR and the transverse extent of PaS, similar to what was observed following injections in rostral A30. The distribution in PrS and MEC however differed from that resulting from injections in rostral A30 in that labeling was present in proximal PrS and more lateral parts of MEC.

## Injections in caudal A29

Caudal A29-injections (n=35) resulted in comparable projection patterns in PHR. Labeled fibers were mostly located in the dorsal half of PrS, and the dorsal two-thirds of layers V-VI of PaS and MEC. However, in some experiments labeled fibers were also observed in ventral MEC and dorsal LEC. Compared to injections in caudal A30, injections in caudal A29 resulted in labeling also in more proximal parts of dorsal PrS. Additionally, after injections in caudal A29, very few fibers were observed in deep layers of PaS, while a dense patch of fibers was usually observed in MEC. The densest projection to PHR usually targeted the deep layers of dorsal PrS and intermediate dorsoventral MEC. We further observed in some cases single labeled fibers in layer III of MEC (n=13), in POR (n=22) and in HF (n=13).

In a representative animal (18576; P15) BDA was injected in layers I-VI of caudal A29 (*Figure 3*). Labeled fibers left the injection site and traveled through layer VI caudally and ventrally towards the RSC-PrS border. Single fibers penetrated the lamina dissecans and branched in layers I and III of

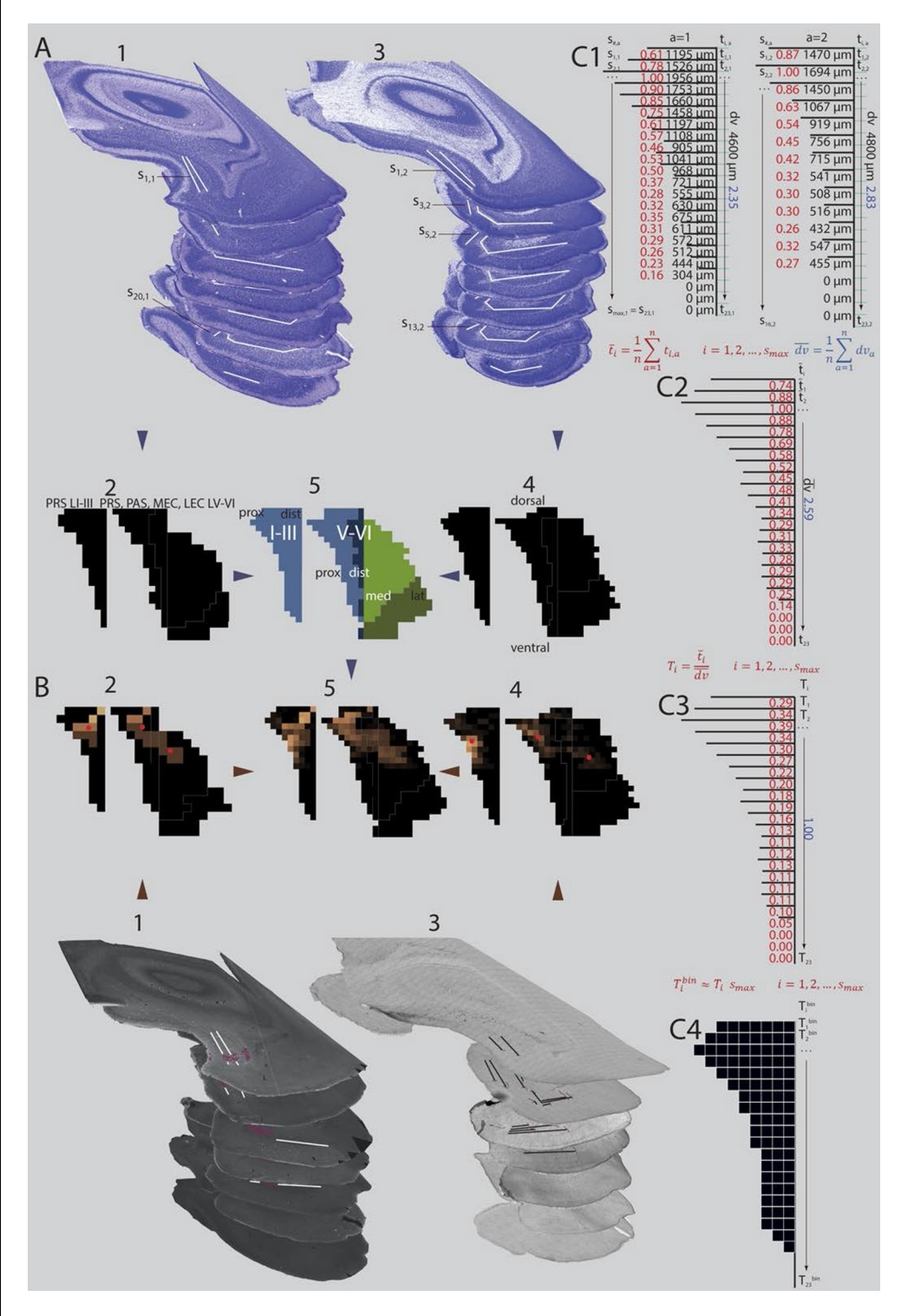

**Figure 4.** Standardized representation of location of labeled axons in PHR. (**A**) Extents of PrS, deep layers of PaS and deep layers of EC were measured in the horizontal plane (**1**; Nissl stained sections from case 18427, **3**; Nissl stained sections from case 18589). In every section, the extents of PrS, PaS, MEC and LEC along the transverse axis were measured. Based on the measurements of all sections in all brains (2 and 4) we binned the PHR along the dorsoventral and transverse axis and made an average representation of PHR (**5**; left: layers I and III of PrS (light blue), right: layers V-VI of PrS (light blue), PaS (dark blue), MEC (light green) and LEC (dark green). Dorsal, ventral, proximal (prox), distal (dist), medial (med), lateral (lat) indicates dorsoventral and transverse axis of the

*Figure 4 continued on next page*

*Figure 4 continued*

flatmaps. $s_{\#,a}$ refers to individual measurements of layers I and III of PrS in C1. (B) Locations of labeled fibers were obtained by measuring the distance between the plexus and the borders of the field in which the plexus was located (1; case 18427, note that we inserted magenta labeled structures to illustrate labeled fibers and their respective positions on the flatmaps). The measurements were performed in every section containing a plexus. The bins between the boundaries of each plexus were given a value ranging from 1 to 3 reflecting weak to dense labeling respectively (color coded in 2, 1 = brown; 2 = orange; 3 = bright yellow). Bins outside the plexus boundaries were given the value 0 (black, no plexus). In experiments in which we observed single labeled axons or a sparse plexus, we measured the distance from each labeled axon to one of the borders of the field in which the plexus was located (3; case 18589; repeating patterns in top sections are artefacts due to stitching of digitized images). We inserted magenta labeled fibers and their respective positions on the flatmaps). We gave each bin in the flatmap a value corresponding to the number of labeled single axons (4; bins not containing any labeled fibers: black; bins with the highest number of labeled axons: bright yellow). For all flatmaps, the centers of mass for layers I and III in PrS, layers V-VI of PrS and PaS combined and layers V-VI of MEC and LEC combined were calculated (red dots). To compare groups of injections, flatmaps of individual injections were normalized to the highest valued bin, transformed to the average flatmap and added together (5; see methods for further details). (C) Binning of layers I and III of PrS along the dorsoventral and along the transverse axis. The example is based on animals 18427 (left) and 18589 (right; both shown *Figure 4A and B*). All subdivisions are binned according to the same algorithm. First the transverse measurements ($s_{\#,a}$, visualized by black lines in C1) and the dorsoventral measurement ($dv_a$) in all sections in all animals were normalized to the longest measurement of the respective subdivision in the particular animal (red (transverse) and blue (dorsoventral) numbers in C1). Next, we binned the dorsoventral axis of each PHR subdivision (represented by turquois lines in C1) in the same amounts of bins as the maximum number of sections containing PHR in a single series ($s_{max}$; 23 in the example) and calculated the transverse extends of each row of bins in each animal ($t_{i,a}$, numbers not shown). Thereafter, we calculated the means of the normalized transverse measurements across all animals for each bin ($\bar{t}i$, red numbers in C2), and the mean across all animals of the normalized dorsoventral extend of PHR ($\overline{dv}$; blue number in C2). The ratio of the mean transverse extends and the mean dorsoventral extend was calculated such that each dorsoventral level was expressed as a value relative to the dorsoventral extent of PHR ($T_i$; red numbers in C3). Finally, the number of bins along the transverse axis for each dorsoventral level ($T_i^{bin}$) was calculated (C4).

dorsal PrS. Single fibers continued into deep layers of PaS and into all layers of POR close to the PaS border. A few fibers were also present in layer III of PaS. The density of labeled fibers increased at more ventral levels and reached its maximum in the middle of the dorsal half of PHR where proximal PrS layers I-III, layers V-VI of PrS were covered by labeled fibers. A few fibers were also labeled in layers V-VI of dorsomedial MEC. Single fibers invaded SUB, layer III of PaS and layer III of MEC. We observed a change in the density and position of labeled fibers along the dorsoventral extent of dorsal PHR. The density of labeling in PHR decreased at successively more ventral levels, while the position of labeled fibers in deep layers shifted from PrS and PaS at dorsal levels towards MEC at intermediate dorsoventral levels. In the ventral third of MEC labeling was only located in lateral parts of MEC. A few labeled fibers were observed in LEC.

All injections in caudal A29 thus resulted in labeling in PHR, comparable to what was seen in case of injections in rostral A29. However, the labeling from caudal A29 extended more ventrally in PHR.

## Specific projection patterns from different parts of RSC

The different labeling patterns in PHR, observed after injections in different parts of RSC indicate that RSC is heterogeneous with respect to the terminal distribution of PHR projections. To systematically analyze this, we measured, in each experiment, the location of the labeled plexus in PHR. Thereafter, we produced normalized flatmaps of the location of the labeled plexus in each experiment (see methods for details). These flatmaps consisted of multiple bins spanning the dorsoventral and transverse axis of PHR (*Figure 4A*). Each bin was assigned a value between 0 (black) and 1 (yellow) reflecting the density of labeled fibers in PHR (*Figure 4B*). This approach allowed us to compare flatmaps across animals of different ages and pool flatmaps of groups of interest. For statistical analysis, we calculated the center of mass of the labeled fields and tested the relationships between this measure and the coordinates of the injection and the age of the animal.

We first performed a cluster analysis to investigate whether the patterns of labeling in PHR were clustered dependent on the location of the injection site or the age of the animal. Using the

distribution patterns as represented in the flatmaps (see methods for details), we identified five clusters. However, the injection sites in RSC associated with each of the clusters were not clustered in RSC such that several of the injections associated with one cluster of distribution patterns partly overlapped with injection sites associated with other labeling clusters in PHR. None of the clusters of labeling in PHR was associated with distinct age groups of animals. We therefore concluded that RSC does not contain regions having distinct projection patterns to PHR, but more likely has a continuous topographical organization of projections to PHR. Therefore, we next assessed whether the labeling patterns in PHC changed systematically in relationship to the rostrocaudal and/or the dorsoventral position of the injections in RSC. For the analysis of the projection patterns, we initially subdivided the injections into two groups depending on whether the injections were located in A29 (n=50) or A30 (n=55). We also subdivided each of the two areas into four equally sized rostrocaudal regions; the rostral quarter of RSC (n=16), intermediate-rostral quarter of RSC (n=22), intermediate-caudal quarter of RSC (n=48) and caudal quarter of RSC (n=19).

To evaluate the projection patterns we analyzed the data in two different ways. First, we pooled the projection patterns of all experiments in each of the eight injection groups described above. We subsequently analyzed the 'mean' projection pattern of each group, as represented on flatmaps of PHR (*Figure 5—figure supplement 1A*). However, the pooled flatmaps are sensitive to the number of bins covered by the labeled axons in each experiment. Experiments in young animals with single labeled fibers in PHR or experiments with small injections resulting in few labeled axons in PHR will contribute fewer 'labeled bins' to the pooled flatmaps, and the pooled flatmaps might therefore be biased towards the experiments with many bins containing labeled axons. Therefore, we also calculated, for each experiment, the center of mass of the projections to respectively layers I and III of PrS, layers V-VI of PrS and PaS combined, and layers V-VI of MEC and LEC combined. The centers of mass served as a quantifiable measure of the location of the labeled axons in PHR, which is independent of the number of fibers labeled in each experiment.

After injections in the rostral part of A29 and A30, labeled fibers were only seen in the dorsal one-third of PHR. Labeling was mainly present in superficial and deep layers of PrS with approximately equal densities, while only a moderate number of labeled fibers was observed in deep layers of PaS and dorsomedial MEC (*Figure 5—figure supplement 1A*). Injections in the three more caudal subdivisions of A29 and A30 resulted in labeling in PHR at successively more ventral levels. Labeling was seen in layers I, III and V-VI of the dorsal half of PrS, in layers V-VI of the dorsal half of PaS and of the medial part of MEC (*Figure 5—figure supplement 1A*). When comparing injections in A29 and A30, we did not see a systematic relationship between the placement of the injection and the dorsoventral distribution of the labeling (*Figure 5—figure supplement 1A*). The results thus suggest that the rostrocaudal placement of injection in RSC is related to the dorsoventral location of the labeled plexus in PrS, PaS, and MEC.

Multiple regressions confirmed a relationship between the rostrocaudal placement of the injection and the dorsoventral location of the center of mass of the labeled axon terminals. More rostrally placed injections resulted in centers of mass of the labeled plexus that were located more dorsally in layers I and III of PrS (*Figure 5A* and *Figure 5—figure supplement 1B*, $\beta=-0.201$, 95% confidence interval (CI) $[-0.290, -0.111]$, $t_{99}=-4.452$, $p<0.001$). The dorsoventral placement of injections was not significantly related to the dorsoventral location of the centers of mass of the projections ($\beta=-0.052$, 95% CI $[-0.144, 0.040]$, $t_{99}=-1.120$, $p=0.265$). There was no significant interaction effect of rostrocaudal-by-dorsoventral placement of the injection on the resulting patterns of anterograde labeling in PHR.

Regarding terminal labeling in layers V-VI of PrS and PaS, more rostrally placed injections in RSC resulted in terminal centers of mass located more dorsally in layers V-VI of PrS and PaS (*Figure 5A* and *Figure 5—figure supplement 1D*, $\beta=-0.133$, 95% CI $[-0.203, -0.063]$, $t_{99}=-3.784$, $p<0.001$). The dorsoventral placement of the injection showed no significant relationship with the dorsoventral location of the centers of mass of the labeling ($\beta=0.020$, 95% CI $[-0.052, 0.091]$, $t_{99}=0.547$, $p=0.585$). There was no significant interaction effect of rostrocaudal-by-dorsoventral placement of the injection.

In layers V-VI of MEC and LEC, more rostrally placed injections resulted in terminal centers of mass located more dorsally in layers V-VI of MEC and LEC (*Figure 5A*, *Figure 5—figure supplement 1F*, $\beta=-0.186$, 95% CI $[-0.269, -0.104]$, $t_{85}=-4.498$, $p<0.001$). The dorsoventral placement of the injection showed a weaker, but significant relationship with the dorsoventral location of the centers

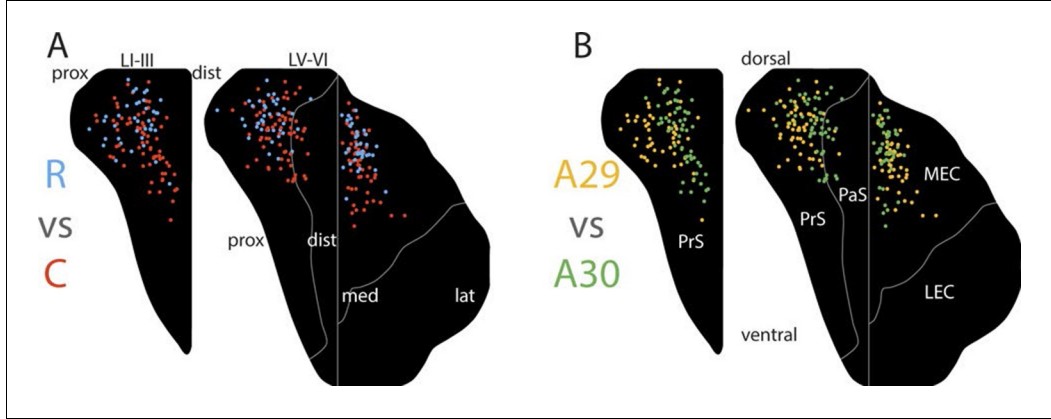

**Figure 5.** Topographical organization of projections. The position of the centers of mass of labelling in layers I and III of PrS, layers V-VI of PrS and PaS and layers V-VI of MEC and LEC is plotted. Each dot is color coded with respect to the rostrocaudal (**A**) or dorsoventral position (**B**) of the injections in RSC (rostral half; blue, caudal half; red, A29; yellow, A30; green). In layers I and III of PrS more caudally placed injections result in projections located more ventral compared to rostrally placed injections (**A**; p<0.001), while ventrally placed injections result in projections located more proximal compared to dorsally placed injections (**B**; p<0.001). In layers V-VI of PrS and PaS caudally placed injections result in projections located more ventral compared with rostrally placed injections (**A**; p<0.001), while ventral and rostrally placed injections result in projections located significantly more proximal compared with dorsally and caudally placed injections (**B**; ventral: p<0.001, caudal: p=0.033). In MEC and LEC layers V-VI, caudal and ventrally placed injections result in projections located significantly more ventral compared to rostral and dorsally placed injections (**A**; caudal: p<0.001, ventral: p=0.032), while ventrally placed injections result in projections located significantly more lateral compared to dorsally placed injections (**B**; p=0.020). Multiple regression was used for all statistical tests (**Figure 5—source data 1–12**). For flatmaps of the projection patterns see **Figure 5—figure supplement 1**.

The following source data and figure supplement are available for figure 5:

**Source data 1.** Datapoints used in multiple regressions.
**Source data 2.** Datapoints used in multiple regressions.
**Source data 3.** Datapoints used in multiple regressions.
**Source data 4.** Datapoints used in multiple regressions.
**Source data 5.** Datapoints used in multiple regressions.
**Source data 6.** Datapoints used in multiple regressions.
**Source data 7.** Datapoints used in multiple regressions.
**Source data 8.** Datapoints used in multiple regressions.
**Source data 9.** Datapoints used in multiple regressions.
**Source data 10.** Datapoints used in multiple regressions.
**Source data 11.** Datapoints used in multiple regressions.
**Source data 12.** Datapoints used in multiple regressions.
**Figure supplement 1.** Topographical organization of RSC-PHR projections in pups.

of mass of the labeling since more ventrally placed injections in RSC had centers of mass located more ventral in PHR (*Figure 5B*, *Figure 5—figure supplement 1F*, β=0.089, 95% CI [0.008, 0.171], $t_{85}$=2.181, p=0.032). There were no significant interaction effects of rostrocaudal-by-dorsoventral placement of the injection. Based on these data we conclude that the rostrocaudal position of injections in RSC determined the dorsoventral location of the labeled plexus in PrS, PaS and MEC.

We subsequently analyzed whether the location of the injection site influenced the transverse position of the labeled axons in the identified PHR subdivisions. A visual analysis of the plotted centers of mass and the pooled flatmaps showed that in layers I and III of PrS the labeling was generally located distally in case of injections in A30 (*Figure 5B*, *Figure 5—figure supplement 1A*). After injections in A29, proximal PrS was also covered by labeled fibers. This suggested that the dorsoventral position of the injection in RSC is related to the transverse position of the labeled fibers in PHR. This suggestion was substantiated through multiple regression analysis, showing that following ventral injections, the centers of mass of the labeled plexus in layers I and III of PrS were significantly more proximal compared to those following injections in dorsal RSC (*Figure 5B* and *Figure 5—figure supplement 1C*; β=0.392, 95% CI [0.267, 0.518], $t_{99}$=6.200, p<0.001). The dorsoventral placement of the injections showed no significant relationship with the dorsoventral location of the centers of mass of the labeling (β=0.117, 95% CI [-0.005, 0.239], $t_{99}$=1.902, p=0.060). There was no significant interaction effect of rostrocaudal-by-dorsoventral placement of the injection.

In layers V-VI of PrS and PaS, the centers of mass were located significantly more proximal after injections in ventral RSC compared to injections in dorsal (*Figure 5B* and *Figure 5—figure supplement 1E*; β=0.292, 95% CI [0.192, 0.392], $t_{99}$=5.780, p<0.001). The rostrocaudal placement of the injection did show a weaker, but significant relationship with the proximodistal location of the labeled fibers (*Figure 5A* and *Figure 5—figure supplement 1E*, (β=0.107, 95% CI [0.009, 0.205], $t_{99}$=2.164, p=0.033); injections in caudal RSC had centers of mass located more distal in PrS layers V-VI, compared to what was seen following injections in rostral RSC. Additionally, we observed a significant rostrocaudal-by-dorsoventral placement of injection interaction effect, since injections located more dorsal and more caudal had centers of mass in significantly more distal parts of layers V and VI of PrS and PaS (β=1.039, 95% CI [0.529, 1.548], $t_{99}$=4.049, p<0.001).

The dense and extensive labeling in layers V-VI of MEC was clearly seen in case of injections in A29, while a more restricted area, medially in MEC, was covered after injections in A30 (*Figure 5—figure supplement 1A*). Multiple regression analysis confirmed that the centers of mass of the terminating axons were located more lateral after injections in ventral RSC compared to after injections in dorsal RSC (*Figure 5B* and *Figure 5—figure supplement 1G*; β=−0.121, 95% CI [−0.222, −0.020], $t_{85}$=−2.378, p=0.020). However, the rostrocaudal placement of the injection had no significant relationship with the mediolateral location of the labeled fibers (β=−0.048, 95% CI [−0.150, 0.054], t=−0.937, p=0.351). There was no significant interaction effect of rostrocaudal-by-dorsoventral placement of the injection.

The overall analysis thus supported the conclusion that the dorsoventral position of the injection in RSC determines the transverse position of the labeled fibers in PrS, PaS, and MEC. By plotting each of the transverse coordinates of the centers of mass against the dorsoventral coordinate of the injection we did not observe any discrete 'jumps' (data not shown). This suggests that the topographical organization of projections from RSC to PHR is not organized into discrete projection patterns from each of A30 or A29 or its subdivisions a, b or c, but rather is organized as a continuous dorsoventral gradient, similar to what has been reported for the adult situation.

## Projection patterns in different age groups

Next, we aimed to investigate how the topographical organization of the projections developed during the postnatal period. We binned the experiments in three age groups, aged P1-P6 (n=33), P7-P13 (n=52), and injections in animals older than P14 (n=20). The labelling patterns from all experiments in each of these age groups were in general similar (*Figure 6—figure supplement 1*, All areas). In all age groups, layers I, III and V-VI of dorsal PrS, layers V-VI of dorsal PaS, and layers V-VI of medial MEC were labeled following injections in RSC. Additionally, for all age groups, the dorsal PrS was the most commonly labeled part of PHR. The distribution of labeling was comparable between all age groups.

We subsequently refined this analysis to study differences between rostral and caudal RSC with respect to labeling patterns in PHR. We plotted the centers of mass of the labeled projections

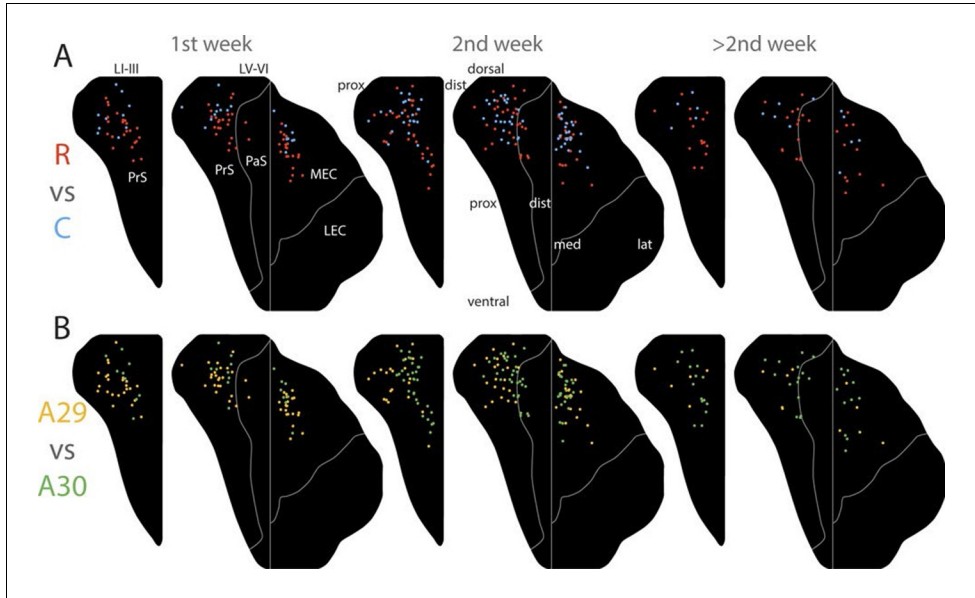

**Figure 6.** Development of topographies. The position of the centers of mass of the labelling in layers I and III of PrS, layers V-VI of PrS and PaS and layers V-VI of MEC and LEC is plotted (see also *Figure 6—figure supplement 2*). Each dot is color coded with respect to the rostrocaudal (**A**) or dorsoventral position (**B**) of the injections in RSC (rostral half; blue, caudal half; red, A29; yellow, A30; green). Left column; animals aged P1-6, middle column; animals aged P7-13, right column; animals aged P14-28. Age is not related to the dorsoventral position of the plexus (PrS LI-III: p=0.876; PRS and PaS LV-VI; p=0.187; EC LV-VI: p=0.198) or the transverse position of the plexus (PrS LI-III: p=0.641; PrS and PaS LV-VI; p=0.325; EC LV-VI: p=0.402). Multiple regression was used for all statistical tests (*Figure 5—source data 1–12*). For flatmaps of the projection patterns see *Figure 6—figure supplement 1*.

The following figure supplements are available for figure 6:

**Figure supplement 1.** Flatmaps of projection patterns of different age groups.

**Figure supplement 2.** Development of topographies.

observed after injections in the rostral half (n=10, 1st week; n=32, 2nd week; n=5, 3rd week and older) and caudal half of RSC (n=23 1st week, n=30 2nd week, n=15 3rd week and older; *Figure 6A*). The overall patterns across the age groups were similar and resembled the data including all injections in each subgroup. After injections in the rostral half of RSC, the labeling patterns in all age groups were limited to layers I, III and V-VI of dorsal PrS, layers V-VI of dorsal PaS and layers V-VI of MEC (*Figure 6—figure supplement 1* R). However, in MEC, animals aged younger than a week had a tendency to only display labeled fibers in the most dorsomedial part while older animals also displayed labeled fibers in more lateral and ventral parts of MEC.

Next, we split each age group in two different subgroups with respect to the location of the injection in A29 or A30, combining data on rostral and caudal RSC. We plotted the centers of mass of the labeling after injections in A30 (n=9 1st week, n=31 2nd week, n=15 3rd week and older) and A29 (n=24 1st week, n=21 2nd week, n=5 3rd week and older, *Figure 6B*). Following injections in A30 in all age groups, the labeled plexus were located in layers I, III and V-VI (*Figure 6—figure supplement 1* A30). In animals aged younger than a week, a tendency for less dense labeling was seen in ventral PHR compared to the older age groups. After injections in A29, all age groups showed comparable labeling patterns. In PrS layers I, III, and V-VI, the terminating axons in all age groups were located more proximally compared to the projection patterns seen after injections in A30 (*Figure 6—figure supplement 1* A29). Similar to injections in A30, in animals aged younger than a week, A29 injections resulted in less labeling in ventral levels of PHR. This effect was most obvious in MEC.

After injections in caudal A29 and A30, the labeling patterns were in general similar across all age groups (*Figure 6—figure supplement 1C*). Injections in caudal RSC resulted in preferred labeling in more ventral parts of PHR compared to injections in rostral RSC.

A descriptive assessment of the projection patterns thus suggested that the different age groups had comparable patterns of labeling. The topography along the transverse axis of PrS and PaS seen after A29 and A30 injections was observed already during the first postnatal week. In addition, in all age groups, caudal injections resulted in more extensive labeling in ventral PHR compared to rostral injections. However, injections in younger animals tended to have less labeled fibers in ventral PHR. To investigate this phenomenon more carefully, we plotted the coordinates of the centers of mass of the axonal labeling as a function of age and the location of the injection (*Figure 6* and *Figure 6—figure supplement 2*). This analysis indicated that already from the earliest postnatal ages, the position of the centers of mass are organized as described above. Already at the earliest ages, we observed that different rostrocaudal levels of RSC project to different dorsoventral levels of PrS and PaS. Even though we observed a tendency for the youngest animals to not display labeling more ventrally in PHR, age had no significant effect on the position of the centers of mass along the dorsoventral axis in neither layers I and III of PrS (*Figure 6—figure supplement 2A*, $\beta$=0.000, 95% CI [−0.005, 0.004], $t_{99}$=−0.157, p=0.876), in layers V-VI of PrS and PaS (*Figure 6—figure supplement 2B*, $\beta$=−0.002, 95% CI [−0.006, 0.001], $t_{99}$=−1.328, p=0.187), nor in layers V-VI of MEC and LEC in older animals (*Figure 6—figure supplement 2C*, $\beta$=−0.003, 95% CI [−0.007, 0.001], $t_{85}$=−1.299, p=0.198). To check if animals aged younger than a week had less labeled fibers in ventral PHR compared to older animals, we converted the continuous age variable to a discrete variable were animals aged younger and older than a week were considered as two different groups. However the age groups were not significantly related to the location of the center of mass (lowest p-value=0.221). In neither of the regression analyses, we observed any significant location-by-age interaction effects.

Already in the youngest cases, the centers of mass in layers I and III of PrS and layers V-VI of PrS and PaS after injections in A29 were located more proximal compared to those resulting from injections in A30 (*Figure 6B* and *Figure 6—figure supplement 2D and E*). Age did not predict the location of the centers of mass along the transverse axis for neither layers I and III of PrS (*Figure 6—figure supplement 2D*, $\beta$=0.001, 95% CI [-0.005, 0.008], $t_{99}$=0.468, p=0.641), layers V-VI of PrS and PaS (*Figure 6—figure supplement 2E*, $\beta$=0.002, 95% CI [-0.002, 0.007], $t_{99}$=0.990, p=0.325), nor layers V-VI of MEC and LEC (*Figure 6—figure supplement 2F*, $\beta$=0.002, 95% CI [-0.003, 0.008], $t_{85}$=0.843, p=0.402). However, for layers V-VI of PrS and PaS there was a significant interaction of rostrocaudal placement of injection-by-age ($\beta$=-0.033, 95% CI [-0.056, -0.11], $t_{99}$=-2.995, p=0.004), indicating that at older ages, respectively more rostral injections resulted in labeling in more distal parts of layers V-VI PrS and PaS.

Based on these results we conclude that the heterogeneous projection patterns observed in later postnatal stages is also present in the first postnatal stages.

## Projection patterns in adolescents and adults

We next asked whether the projection patterns in PHR, observed in young animals, are also present in adult animals (approximately 3 months). To this end, we reanalyzed a dataset described in earlier publications from our lab (*Jones et al., 2005*; *Jones and Witter, 2007*). Analyses of the cases individually, suggested that the organization of the projection pattern as we described in young animals was also present in adult animals. After injections in A30, a dense fiber plexus was observed distally in layers I and III of dorsal PrS (*Figure 7A*). Additional labeled fibers were observed in layers V-VI of distal PrS, PaS and medially in MEC, comparable to the projection pattern we observed after injections in A30 of pups (*Figure 5—figure supplement 1A* and *Figure 6—figure supplement 1*). Compared to injections in A30, injections in A29 resulted in labeled fibers more proximally in PrS and more laterally in MEC (*Figure 7B*). This topographical organization seems to be graded, since the injection located at the border between A29 and A30 displayed labeled fibers in a mediolateral position between the dorsally- and ventrally injected cases (compare *Figure 7C* with A and B respectively). Compared to rostrally placed injections (*Figure 7A and B*), we observed that caudally placed injections displayed more labeled axons in ventral PrS and MEC, and the area receiving the densest projection in both PrS and MEC was shifted more ventrally (*Figure 7C*). Taken together, these findings suggest that the RSC to PHR projections in the postnatal brain are similarly organized to the adult ones.

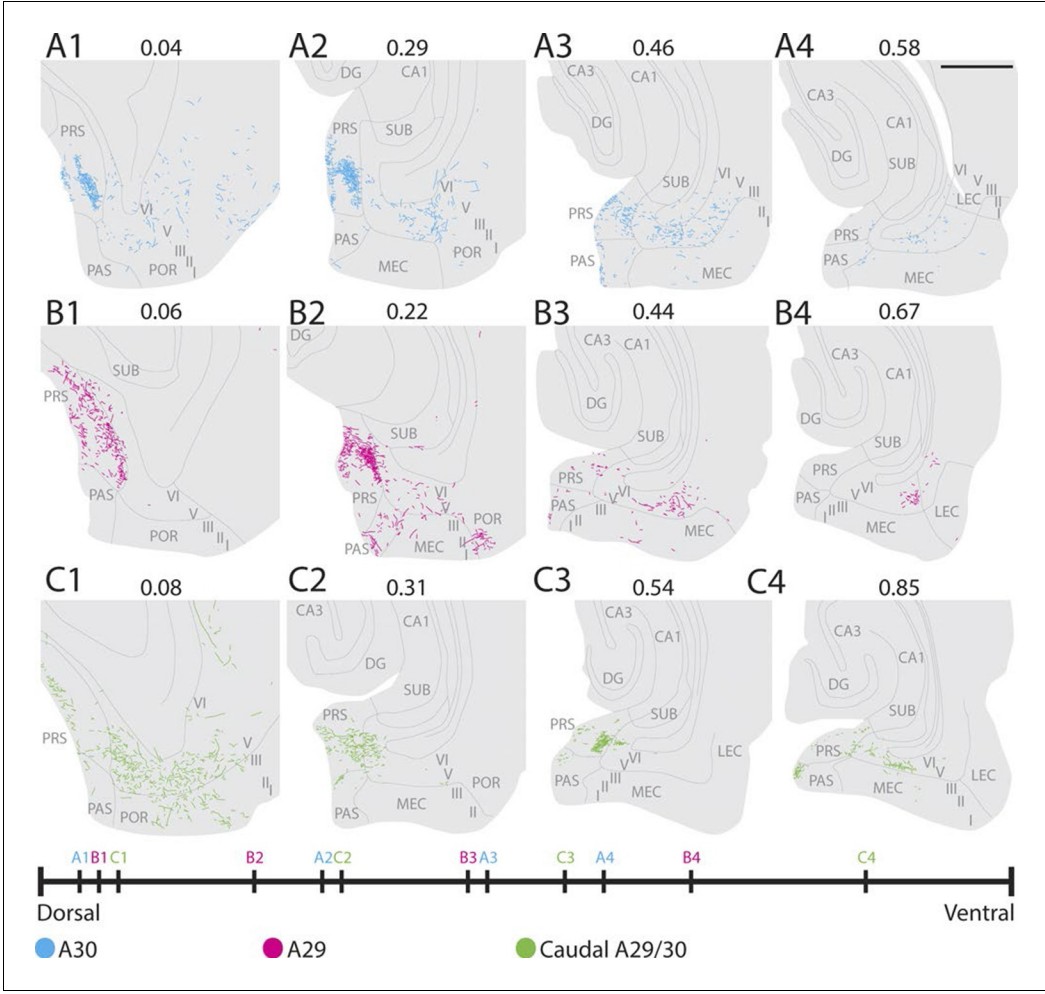

**Figure 7.** Labelling patterns in the adult. (**A**) An injection in intermediate rostrocaudal A30 resulted in dense labelling in layers I and III of distal PrS and a few fibers in deep PrS, PaS and POR (A1). More ventrally, fibers also invaded the MEC with the most fibers located medially in MEC (A3). (**B**) An injection in intermediate rostrocaudal A29 resulted in a dense fiber plexus in layers I and III of proximal PrS and POR (B2). At this dorsoventral level a few fibers also invaded PaS. More ventrally, a fiber plexus was present in lateral MEC. (**C**) An injection at the border of A29 and A30 in caudal RSC resulted in fibers located in layer I of PrS and deep PaS and POR dorsally (C1). More ventrally, a dense projection to layers I and III of PrS (C2 and 3) and a moderate projection to intermediate mediolateral MEC was present (C4). In A-C numbers above each section depict normalized dorsoventral position of the section and the line at the bottom of the figure represents the relative dorsoventral position of each section. Scale bar equals 1000 µm.

## Development of density of projection

We aimed to investigate whether there was a chronological development of the number of RSC-axons which could be traced within PHR. In the first postnatal week, very few fibers were labeled in PHR. At P1-P3 we only observed single unbranched fibers in PHR (*Figure 8A–D*). Along the fibers, axonal growth cone-like swellings were observed, independent of whether the fibers were located in the cortex or in the white matter. At P3, we observed the first branching fibers in PHR, although these fibers typically branched only once or twice in each section. After P3, the plexus gradually increased in density and complexity. However, we did not observe an adult-like plexus until P12 (*Figure 8F*). At this age, we observed dense plexus, with fibers displaying numerous branching points and each fiber containing fine extensions and protrusions, similar to what was seen in older animals and adults (*Figure 8G*). Based on these observations we concluded that the number of RSC fibers in PHR increase gradually from single fibers at P1 to adult-like densities around P12.

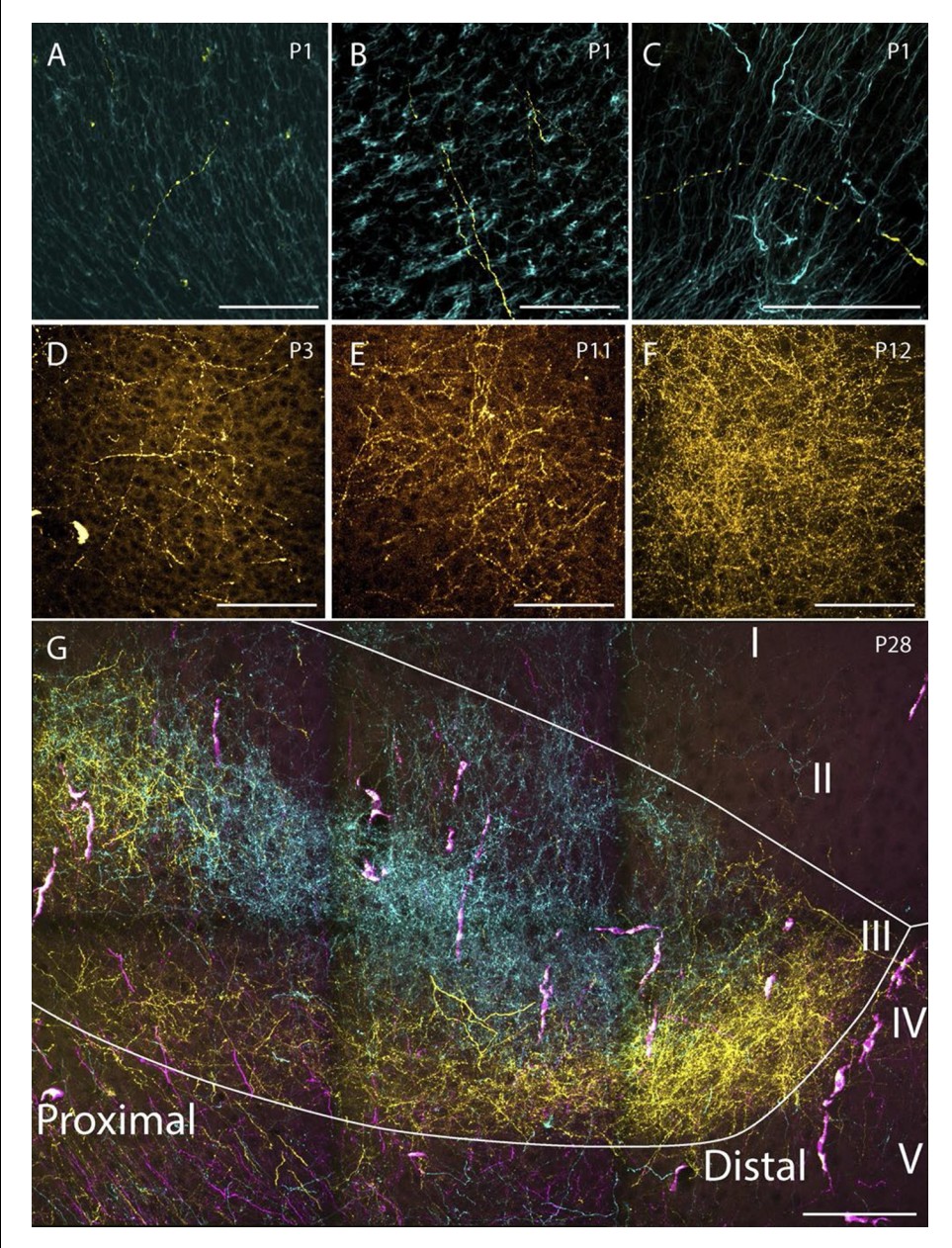

**Figure 8.** Examples of the densest fiber plexus observed in PHR at different ages. At birth, single unbranching fibers were present in superficial layers of PrS (A), deep layers of PrS (B) and deep layers of MEC (C). At P3, some branching fibers were observed (D, PrS). These fibers typically branched only once or twice. At P11 (E, PrS) the complexity of the fiber plexus increased as fibers have multiple branching points and thin fibers are seen within the plexus. At P12 (F, PrS) the first plexus which was comparable to plexus in adolescent animals (G, PrS) and adults (data not shown) was observed. In the experiment shown in G, the terminal distribution of three differentially labeled projections is illustrated. A plexus resulting from an injection in intermediate-rostral A29 (yellow) terminated throughout layer III of PrS, while the plexus observed after an injection in intermediate-rostral A30 terminated in the center of layer III of PrS (cyan). Fibers originating from a third injections in intermediate-caudal RSC (magenta) are observed, however the densest fiber plexus is located in a section at more ventral levels of PrS. The large, overlapping magenta and yellow structures are endothelium which take up alexa-preconjugated dextran amines. White line depicts the deep and superficial borders of layer III. Scale bars equal 100 µm.

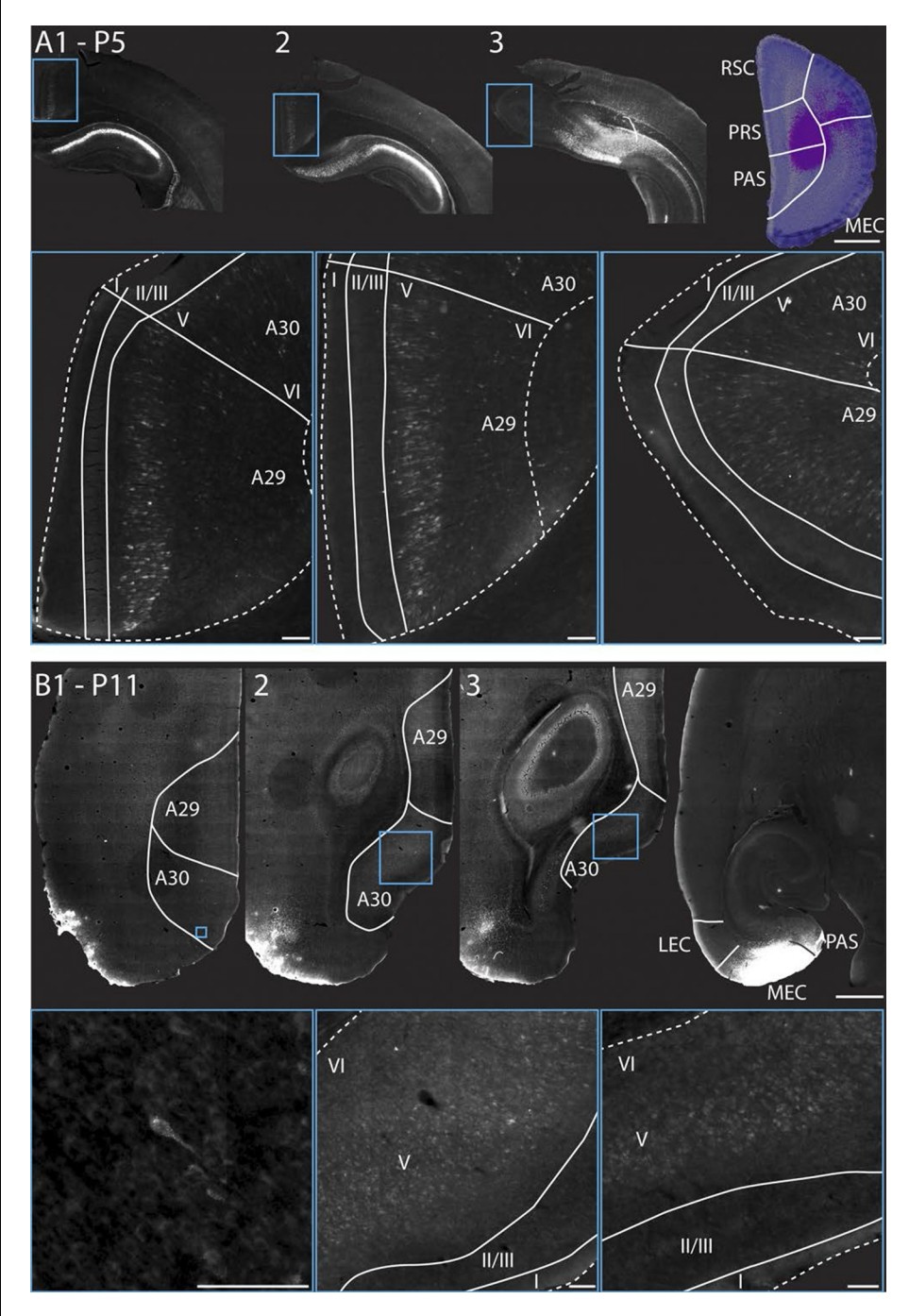

**Figure 9.** RSC projections to PHR arise from neurons in layer V. (**A**) A FB injection in deep layers of mainly PrS and PaS (top right; P5, coronal section) resulted in retrograde labeling of neurons in CA1, SUB and RSC (top row). Coronal sections shown are from intermediate-rostral RSC (1), intermediate-caudal RSC (2) and caudal RSC (3). Cyan squares depict size and position of high-power images in the bottom row. Bottom row: High power images of retrogradely labeled neurons in superficial layer V of RSC. Dashed lines depict the pia and the border between cortex, white matter and the corpus callosum. Solid lines depict borders between layer I, layers II-III and layer V and border between A29 and A30. Scale bars equal 100 μm (high power images) and 1000 μm (low power images). (**B**) FB was injected in an intermediate dorsoventral level in MEC and PaS (top right; P11, horizontal sections) and resulted in retrogradely labeled neurons in superficial layer V of caudal RSC (top row). Horizontal sections are organized from dorsal (1) to ventral (3). Cyan squares depict location of high-power images in the bottom row. White lines depicts borders of RSC and border between A29 and A30. Bottom row: High power

*Figure 9 continued on next page*

*Figure 9 continued*

images of retrogradely labeled neurons in superficial layer V of RSC. Dashed lines depict the pia and the border between cortex and white matter. Solid lines depict borders between layer I, layers II-III and layer V. Scale bars equal 100 µm (high power images) and 1000 µm (low power images).

## RSC neurons projecting to PHR

Since most of the RSC-PHR projections seemed to be fully developed very early during postnatal development, we aimed to investigate whether the PHR projecting neurons in RSC were located in layer V, similar to adults (*Burwell and Amaral, 1998a*). After fast blue injections in PHR of pups we identified labeled neurons in RSC in superficial layer V at P5 (*Figure 9A*) and P11 (*Figure 9B*), which suggested that the RSC-PHR projections are adult-like with respect to the layer in which the neurons are located already during the first postnatal week of development. These observations are in line with our anterograde material in which we observed that all injections that did not cover parts of layer V (n=8) did not result in any labeled axons in PHR.

## Discussion

To study the development of RSC projections to HF-PHR, we injected anterograde tracers in RSC of rats aged P0-28. We conclude that the postnatal RSC projects densely to all layers of PrS and posterior POR and deep layers of PaS and MEC and weakly to deep layers of LEC and to SUB. Our retrograde experiments showed that the origin of these projections were neurons located in superficial parts of layer V of RSC. These findings are in accordance with previous work in the adult (*Wyss and Van Groen, 1992*; *Shibata, 1994*; *Burwell and Amaral, 1998a*; *Jones and Witter, 2007*). Additionally, we report that the RSC projections to PrS, PaS and EC are topographically organized similarly already in the youngest postnatal rats and in adult rats (*Figure 10*). The notion that rostral RSC only projects to dorsal PHR, while caudal RSC projects to additional more ventral parts of PHR is in accordance with previous work in the adult (*Wyss and Van Groen, 1992*; *Shibata, 1994*; *Jones and Witter, 2007*). Second, dorsal RSC (A30) projects preferentially to distal PrS, PaS and medial MEC, while more ventral parts of RSC (A29) project significantly more to proximal PrS and more lateral parts of MEC. To our knowledge, such topographies have not been reported in earlier studies.

The observation that A30 is preferentially connected to distal PrS, PaS and medial MEC, while A29 is preferentially connected to proximal PrS and more lateral parts of MEC, but not PaS is in line with other connectional and functional differences. The PrS to MEC projection is topographically organized such that distal PrS projects to medial MEC, while proximal PrS projects to more lateral parts of MEC (*Shipley, 1975*; *Honda and Ishizuka, 2004*). Furthermore, these partner domains in PrS and MEC are selectively innervated by different areas of SUB along its transverse axis such that distal SUB projects to distal PrS and medial MEC, while more proximal parts of SUB, with the exclusion of the very proximal part, project to proximal PrS and more lateral MEC (*Witter, 2006*; *O'Reilly et al., 2013*). This indicates the existence of a connectional route linking A30, medial MEC, distal SUB and distal PrS to each other. A parallel route links A29 with more lateral MEC, more proximal SUB and proximal PrS. Although no clear transverse gradients in spatial modulation have been reported in MEC, electrophysiological properties of neurons show a transverse gradient, indicative that more medially positioned grid cells might be more precisely spatially tuned (*Canto and Witter, 2012*). Furthermore, neurons in distal SUB are more spatially modulated compared to those in proximal SUB (*Sharp and Green, 1994*). All observations thus point to a differentiation of A29 and A30 where A30 is connected preferentially to more spatially modulated neurons in medial MEC and distal SUB compared to A29 which is more connected to neurons in lateral MEC and more proximal SUB.

Additional cortical and subcortical connections are in line with this proposed differentiation between both areas of RSC. The projections to anterior cingulate cortex are differentially organized for A29 and A30 and while A30 is connected to visual area A17 and A18b, A29 is only connected to A18b. With respect to thalamic connections, A29 is connected to the anterodorsal and anteroventral nuclei while A30 preferentially connects with the anteromedial nucleus (*van Groen and Wyss, 1990*, *1992*; *Shibata, 1998*; *van Groen and Wyss, 2003*; *Jones et al., 2005*). Taken together, these connectional differences are in line with reported functional differences between the two areas

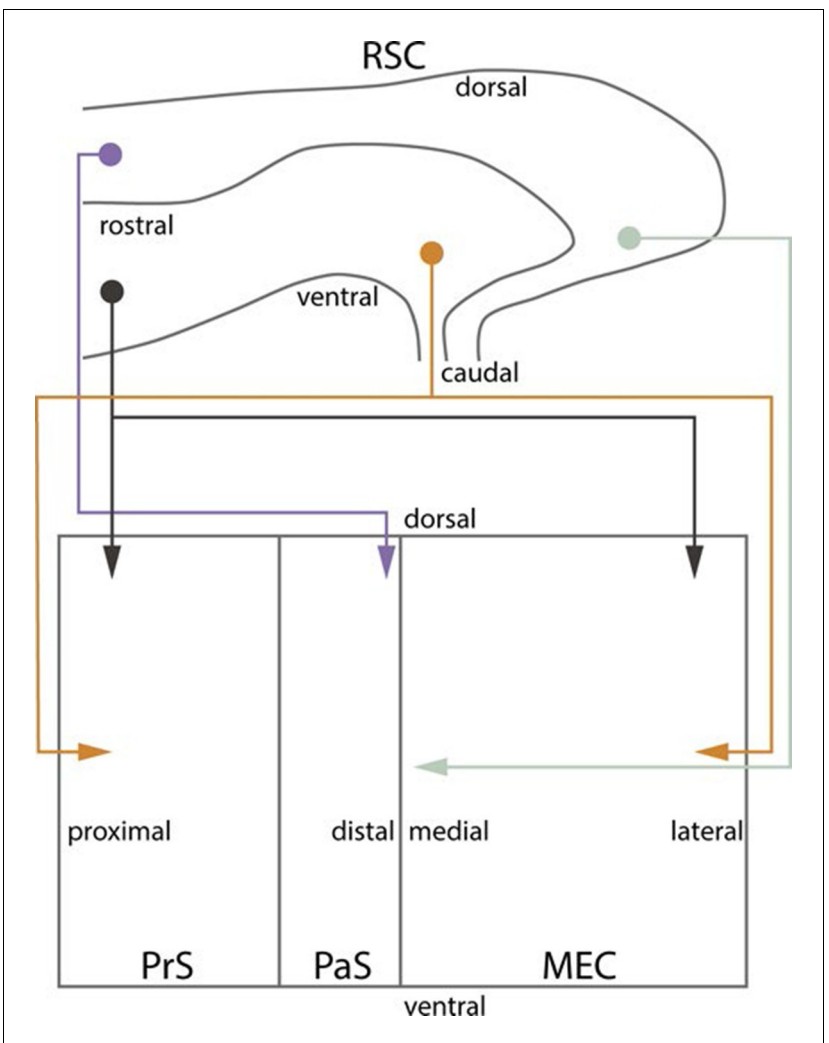

**Figure 10.** Summary of topographical organization of projections from RSC to PHR in the developing and adult brain. Schematic representation of the organization of the projections from RSC (top) to PHR (bottom). Projections from RSC to PHR terminate mainly in PrS, PaS and MEC. Projections originating from rostral RSC (purple and black) terminate in dorsal PHR, while projections originating from caudal RSC (yellow and grey) terminate in more ventral parts of PHR. Projections originating from ventral RSC (black and yellow) terminate in proximal PrS and in lateral parts of MEC, while projections originating from dorsal RSC (purple and grey) terminate in distal PrS and in medial parts of MEC.

(*van Groen et al., 2004*; *Vann and Aggleton, 2005*; *Pothuizen et al., 2009*; *Pothuizen et al., 2010*). Such a functional differentiation between A29 and A30 could thus possibly result in functional differences along the transverse axes of PrS, PaS and MEC.

During the postnatal period we observed an overall increase in the density of labeled axonal branches in all PHR subregions. Even though we did not perform a formal quantification of the number of labeled fibers in PHR, we only observed single unbranching fibers in animals aged P1-2. During the first postnatal week, the number of axons generally increased, and the fibers displayed several branching points towards the end of the first postnatal week. The first terminal plexus with adult-like densities were observed in P12 animals. Even though several other factors, such as tracer type and the number of layer V RSC neurons involved in the injection also had an impact on the number of fibers observed in each experiment, we are confident that the most important predictor of labeled fiber density was the age of the animal. Interestingly, the time window of increased density of RSC afferents in PHR is paralleled by several anatomical and physiological changes in PHR. EC

afferents originating in PaS and PrS become functional from P8 and mature gradually until they are fully adult-like around P14 (*Canto et al., 2011*). Similar developmental timescales have also been reported for the functional development of intralaminar projections within EC (*O'Reilly et al., 2010*).

Our temporal analysis revealed that RSC axons destined for PHR migrate directly into their area of termination and thereafter keeps their position constant while the number of axonal branches and the total axonal spread increases gradually until they reach adult-like plexus features. This observation is supported by our center of mass analyses of the early perinatal RSC to PHR projections, showing that projections originating in different parts of RSC show a striking terminal topography already during the first postnatal week.

Head-direction cells and border cells are all present in PHR when electrodes are lowered into the brain during the second postnatal week (*Bjerknes et al., 2014*; *Bjerknes et al., 2015*), while grid cells mature during the third and fourth postnatal week (*Langston et al., 2010*; *Wills et al., 2010*). Even though no published experiments have investigated whether border cells and head-direction cells are present before the second postnatal week, our data indicate that the topographical organization of RSC terminals in PHR is present before spatially modulated neurons are present in PHR and that adult-like axonal densities can be observed approximately at the same time-point as the first border cells and head-direction cells are observed in PHR. This conclusion is comparable to what has been observed for intrinsic HF-PHR connectivity which is also topographically organized already at early postnatal periods, demonstrating increased plexus densities for intrinsic HF-PHR projections during ongoing development (*Fricke and Cowan, 1977*; *Borrell et al., 1999*; *O'Reilly et al., 2013*; *O'Reilly et al., 2014*). It is also comparable to what has been reported in several other developing brain systems. For instance, for thalamocortical projections in the visual system of both monkeys, cats and ferrets, the first arriving axons in the visual cortex already show a topographical organization into ocular dominance columns (*Horton and Hocking, 1996*; *Crowley and Katz, 2000*; *Crair et al., 2001*). Similarly, the rat ventral posterior thalamic nucleus issue projections to the somatosensory cortex in which the first arriving axons target distinct areas later forming a defined barrel (*Catalano et al., 1996*). Sensory inputs to the olfactory bulb are present long before neurons display a receptive field. Moreover, the development of these inputs is independent of activity in sensory receptors, which suggests that the development of topographies in the olfactory bulb is experience independent (*Lin et al., 2000*). The same conclusion apparently holds for the projections from RSC to PHR, These findings are different from what has been reported for the retinogeniculate projection in several species. In this projection, there is an overshoot of axonal terminals, which are initially diffuse and later pruned into an adult-like topography (*Rakic, 1976*; *Linden et al., 1981*; *Shatz, 1983*; *Godement et al., 1984*). These differences might imply that distinct molecular- or activity based principles governs the axonal termination patterns in different neural projections.

The results presented in this study thus lead us to conclude that the topographical organization of PHR connectivity is present when the first RSC axons arrive in PHR. The densities of the terminal plexus appear to develop gradually without any clear signs of pruning, though it remains to be determined whether connectional reorganization occurs at the synaptic level during development. The first plexus with adult-like densities can be observed around P12 which is around the time when the first spatially modulated neurons in PHR have been observed (P11), but before eye-opening and before the animals starts to navigate (around P15) and thus before adult like grid cells are observed in PHR (after P25; *Langston et al., 2010*; *Wills et al., 2010*; *Bjerknes et al., 2014*; *Bjerknes et al., 2015*). These findings suggest that RSC afferents might be important for the development of head-direction and border cells.

## Materials and methods

### Surgeries and perfusion

Eighty two female and male Long Evans rats aged between P0 and P28 were used in this study. Additionally, we reanalyzed anterograde injections performed in approximately 3 months old adult Wistar rats (*Jones et al., 2005*; *Jones and Witter, 2007*) and retrograde injections performed in Long Evans pups (*O'Reilly et al., 2014*). We refer to these original publications for a detailed description of the experimental protocols.

The pups were bred in-house and housed in enriched cages together with their parents and litter-mates. Cages were checked every morning and evening for pups and the day pups were observed was considered P0. We use the day of perfusion designating the age of the animal. To avoid unnecessary stress for the animals, litters with more than ten pups were culled to ten pups at P0 or P1. At P21, the pups were separated from their parents and moved to cages together with littermates of the same sex. The animals lived in a controlled environment ($22 \pm 1°C$; humidity 60%; lights on from 8:00 P.M. to 8:00 A.M.). Food and water were available *ad libitum*. The experimental protocols followed the European Communities Council Directive and the Norwegian Experiments on Animals Act and local directives of the responsible veterinarian at the Norwegian University of Science and Technology.

All surgeries were conducted under isoflurane gas anesthesia. Animals were placed in an induction chamber and fully anesthetized before they were moved to a stereotaxic frame. The head was fixed using a neonatal mask and mouthpiece (model 973-B; Kopf, Tujunga, CA) and zygoma ear cups (model 921; Kopf). Animals older than P18 were mounted in a small-sized adult mask and the head was fixed with blunted ear bars. Before incision, the skin was disinfected with 2% iodine in 65% ethanol, and a local analgesic bupivacain (0.2 ml per 100 g bodyweight of a 0.5 mg/ml solution; Marcain, Astra Zeneca, London, UK) was injected subcutaneously at the place of incision. The skin was opened with a small-sized and sharp tipped scissor. After incision, the mouthpiece and ear cups were adjusted so that bregma and lambda were aligned horizontally. Before injecting, bone over the place of injection and over the posterior extreme of the sagittal sinus was removed. The exact place of injection was measured using the junction of the transverse- and sagittal sinus as a reference for the anteroposterior coordinate, the lateral edge of the midsagital sinus as a reference for the medio-lateral coordinate and the level of the dura as a reference for the dorsoventral coordinate. Before injection, the dura was punctured, and glass micropipettes with an outer diameter of 20–25 μm were lowered into the brain (30–0044, Harvard Apparatus, Holliston, MA; pulled with a PP-830 puller, Narishige, Japan). The anterograde tracers biotinylated dextran amine (BDA; 5% in phosphate buffer (PB; 0.125M in $H_2O$; pH 7.4), 10000 MW, D1956, Invitrogen, Eugene, OR) or preconjugated dextran amines (all 5% in PB, 10 000 MW; Alexa-488 DA, D22910; Alexa-546 DA, D22911; Alexa-647 DA, D22914; Invitrogen) were iontophoretically injected through the micropipettes into RSC (4–6 μA, alternating currents, 6 s on/6 s off, for 5–15 min, 51595; Stoelting, Wood Dale, IL). Throughout the surgery, appropriate amounts of sterile saline (room temperature) were administered subcutaneously to avoid dehydration. Animals were also administered carprofen during surgery as a post-surgery analgesic (1 ml per 100 g bodyweight of a 0.5 mg/ml solution; Rimadyl, Pfizer, New York, NY). After surgery, the incision was sutured and the pups were allowed to recover under a heating lamp. When fully awake, the animals were returned to maternal care until the time of sacrifice.

We euthanized the animals 18–30 hr after surgery under a terminal anesthesia with isoflurane. The thorax was opened and cold Ringer's solution (8.5 g NaCl, 0.25 g KCl and 0.2 g $NaHCO_3$ per liter of $H_2O$, pH 6.9) was transcardially perfused through the body. When the liver turned pale the perfusion solution was changed to a 4% solution of freshly depolymerized paraformaldehyde in PB (pH 7.4). In case of P0-P2 animals, 0.1% glutaraldehyde was added to the fixative. The brain was removed from the skull and postfixed overnight at 4°C in the same fixative. Twenty-four hours after perfusion, the brains were transferred to PB containing 2% dimethyl sulfoxide (DMSO; VWR, Radnor, PA) and 20% glycerol (VWR).

## Tissue processing

Brains were cut with a freezing microtome (HM-430 Thermo Scientific, Waltham, MA) in 40 or 50 μm thick horizontal sections. Depending on the age of the animal sections were collected in four to six equally spaced series. One of the series was mounted directly on superfrost slides (10149870, Thermo Scientific). The remaining series of sections were collected in vials containing 2% DMSO and 20% glycerol in PB and stored at -20°C until further usage. The mounted series was Nissl-stained by first dehydrating the sections in increasing ethanol solutions (50%, 70%, 80%, 90%, 100%, 100%, 100%) followed by two minutes in xylene (VWR) to clear the sections. Thereafter, the sections were rehydrated in decreasing ethanol solutions (opposite order as the dehydration protocol) and placed in cresyl violet solution for two to six minutes. Subsequently, the sections were rinsed quickly in water and placed in 50% ethanol containing acetic acid to differentiate the staining. The

sections were dehydrated in ethanol, cleared in xylene and finally coverslipped with Entellan (107961, Merck, Darmstadt, Germany).

## Visualization of anterograde tracers

In case of brains with BDA injections, one series of sections was rinsed three times for 10 min in PB and then three times for 10 min in tris(hydroxymethyl)aminomethane (Tris)-buffered saline (50 mM Tris (Merck) and 150 mM NaCl in $H_2O$) containing 2% Triton X-100 (TBS-Tx; Merck, pH 8.0). In experiments were multiple tracers were injected, the sections were incubated with Alexa-conjugated streptavidin (Alexa-405 S32351, Alexa-488, S11223; Alexa-546, S11225; Alexa-633, S21375, Invitrogen) in a 1:200 solution with TBS-Tx overnight at 4°C. In experiments were BDA was the only tracer injected in the brain, sections were incubated for 90 min in TBS-Tx with avidin-biotin-peroxidase (Vectastain Standard PK-4000 ABC kit; Vector, Burlingame, CA) according to the manufacturer's instructions. Subsequently, sections were rinsed three times for 10 min in TBS-Tx and two times for 5 min in Tris-HCl (50 mM Tris in $H_2O$, pH adjusted to 7.6 by adding HCl) and incubated for approximately 15 min in a diaminobenzidine tetrahydrochloride (DAB)–peroxidase solution containing 5 mg DAB (D5905, Sigma-Aldrich, St. Louis, MO) and 3.3 µl $H_2O_2$ (H1009, Sigma-Aldrich) in 10 ml Tris-HCl. Irrespective of whether the incubations were carried out with Alexa-conjugated streptavidin or DAB, the sections were rinsed two times for 5 min in Tris-HCl, and subsequently mounted on glass slides from a 0.2% gelatin solution in Tris-HCl. After overnight drying, they were cleared in toluene and coverslipped with Entellan (Merck).

Sections were inspected with fluorescence illumination at the appropriate excitation wavelength or conventional brightfield illumination (Zeiss Axio Imager M1/2). Digital images of successful injections and anterogradely labeled plexus in HF-PHR were obtained using a slide scanner equipped for either brightfield or fluorescent imaging (Zeiss Mirax Midi; objective 20X; NA 0.8). For illustrative purposes, images of labeled cells and Nissl stained tissue were exported using Panoramic Viewer software (3DHistech, Budapest, Hungary) and processed in Adobe Photoshop and Illustrator (CS6, Adobe Systems, San Jose, CA).

## Assessments of experiments

After the sections were digitized, we aimed to obtain, for each experiment, realistic estimates of the location of the anterogradely labeled axons in PHR and of the locations of the injections in RSC. To achieve this, we first produced an average flatmap of PHR based on measurements of each PHR subdivision in all animals (*Figure 4A*). Second, we measured the location of labeled fibers within PHR and represented the position of the labeled fibers on the average flatmap (*Figure 4B*). Third, we plotted the injections in a reference atlas brain (*Figure 1*).

In order to create an average flatmap, we produced individual flatmaps of all animals. To this end, we delineated all subdivisions within PHR using cytoarchitectonic differences between the different subdivisions (*Boccara et al., 2015*). Borders were established in fluorescent- or DAB-stained sections overlaid with the neighboring Nissl-stained section. In all sections, we measured (using Panoramic Viewer software, 3DHistech, Budapest, Hungary) the extent, along the transverse axis, of subdivisions containing labeled RSC axons, i.e. superficial layers of PrS, deep layers of PrS, deep layers of PaS, deep layers of MEC and deep layers of LEC (*Figure 4A1–4*). The transverse measurements where obtained from all sections of all brains, stored in excel files and the data were further processed using MatLab software (R2015b, MathWorks, Natick, MA).

We next aimed to make one average flatmap, constituting of square bins, representing the mean 'shape' of PHR across all brains. Since brains of animals of different ages have different sizes, we first converted the absolute measurements into normalized values. For this, we divided each measurement by the maximum measured extent of the respective subdivision for the particular animal (*Figure 4C1*). Next, we binned the dorsoventral axis of each PHR subdivision in 29 equally sized bins, since the maximum number of sections containing PHR in a single series was 29. Subsequently, we calculated the mean of the normalized transverse measurements, across all animals, for each of the 29 dorsoventral levels (*Figure 4C2*). This procedure was repeated for each subdivision. To obtain bins each representing a square area of the brain, we first calculated the ratio between the total dorsoventral extent of PHR and the maximum measured extent along the transverse axis of each animal (*Figure 4C1*). Thereafter, we calculated the mean of these ratios across all animals (*Figure 4C2*).

Next, the 29 normalized transverse measurements were divided by the mean of the dorsoventral extents (*Figure 4C3*). This procedure was repeated for each subdivision. This procedure thus resulted in, for each subdivision, 29 dorsoventral levels with different transverse extents, expressed as a value relative to the dorsoventral length of PHR. The transverse extents were subsequently turned into square bins (calculated as the average extent along the transverse axis x 29 and rounded to the nearest integer, *Figure 4C4*) so that each subdivision could be represented as a flatmap containing 29 rows of bins with a variable amount of square bins in each row (*Figure 4A5*).

## Location of labeled fibers

We represented the location of labeled axons in PHR for each experiment within the average flatmap in two ways. In experiments where a dense labeled plexus was present, the distances between the boundaries of the plexus and the borders of the respective PHR subdivisions were measured along the transverse axis (*Figure 4B1*). We did this for each section containing a labeled plexus. The density of labeling in each plexus in each section was subsequently given a value from 1–3 depending on a subjective evaluation. The densest plexus in each experiment was given a value of '3', while weaker plexus were valued '1' or '2' depending on the density relative to the densest plexus observed in the experiment. Alternatively, in experiments in which we observed only single labeled axons or a sparsely labeled plexus, the transverse measurements were obtained as follows. In PrS, the distance from each labeled axon to the proximal border of PrS was measured, while in deep layers of PaS the distance from each labeled axon to the distal border of PaS was measured. In deep layers of MEC and LEC, the distance from each labeled axon to the medial border of the respective subdivision was measured (*Figure 4B3*). In cases of missing or damaged sections, we estimated the putative projection pattern in the section by using the average projection pattern of the sections directly above and below.

To identify the bin(s) in the average flatmap representing the location of labeled fibers, their absolute position, as established above, was normalized within its respective area. In cases where we observed a dense plexus, each bin was given the value corresponding to what was described above (*Figure 4B2*). In cases where we measured the location of labeled fibers, each bin was given a value equal to the numbers of labeled axons present in the location represented by the respective bin (*Figure 4B4*). All bins not containing any labeled axons or located outside the labeled plexus were given the value 0.

To be able to directly compare or pool flatmaps representing either single fibers or dense plexus, we normalized the values of all bins in each experiment to the maximum valued bin in the respective experiment, resulting in bins in each experiment with values ranging between 0, representing no plexus or fibers, or 1, representing the bin with the densest labeling. To compare different sources of variability for the projection patterns, we organized the flatmaps into different groups of interest. For each group, we summed the values of bins representing the same location in all flatmaps (*Figure 4B5*). Finally, we normalized the 'summed' flatmap to the bin with the highest value, similar to the procedure described above. In flatmaps representing projection patterns of several subgroups, normalized flatmaps were summed, so that each subgroup had similar impact on the summed flatmap. For illustrative purposes, the flatmaps of individual experiments or groups of experiments were plotted using MatLab.

In all experiments, we calculated the coordinates for the 'center of mass' of the projections along the transverse and dorsoventral axis for respectively layers I and III of PrS, layers V-VI of PrS and PaS combined and for the combined layers V-VI of MEC and LEC. The center of mass-values of the axonal plexus were used for statistical analyses. In scatter plots of center of mass-values, we defined the extremes of the color scale as ± 2 standard deviations from the mean value. All values between these extremes were plotted using a linear color scale while more extreme values were thresholded to the extreme colors.

## Location of injections

Since brains of different ages have different sizes, we aimed to normalize the position of the injections. In adult rats, RSC can be subdivided into four different cytoarchitectonic subdivisions A30 and A29a, b and c. In the adult brain, these four subdivisions are positioned along the dorsoventral extend of RSC. In the immature cortex, only the cytoarchitectonic border between A29 and A30 is

clearly identifiable. We therefore chose to define the dorsoventral location of the injections in A29 and A30 as a continuous coordinate. This measure is thus indirectly related to the classical discrete cytoarchitectonic subdivisions. We used the recently released reference 3D-atlas brain (*Papp et al., 2014*; *2015*) to map all injections in a standardized space, irrespective of age. First, we identified coordinates of the dorsal, ventral, rostral and caudal border of respectively A29 and A30 in the atlas brain. The lines between the respective coordinates were smoothed using local regression. Next, we calculated the cutting angles of our experimental brains relative to the atlas brain and made sections of the standard atlas brain with the same cutting angles. We identified atlas-sections containing landmarks and cytoarchitectonic borders present in the section containing the center of each injection. The atlas coordinate of the center of each injection was recorded and served as an age-normalized 3D point-measure of the injection location within RSC. For illustrative purposes, the coordinates of all injections were plotted within the 3D volume (*Figure 1A*, ITK-SNAP, NIH).

Since the caudal RSC cortex is curved both along the dorsoventral and rostrocaudal axis we flattened RSC and transposed each injection onto a 2D plane. This was done by dividing the surface area of A29 and A30 in the atlas brain into multiple triangles (*Figure 1B*). The coordinates of the dorsal and ventral borders of A29 and A30 determined the coordinates of each triangle. For each injection, we calculated the shortest vector between the injection and the cortical surface within any of the triangles (*Figure 1C*). Thereafter, we calculated the intersection of the vector and the plane within the triangle. This coordinate represented the "transposed" location onto the cortical surface of each injection (*Figure 1B and C*). The normalized 2D coordinate of each injection was defined as follows (*Figure 1C*); the dorsoventral coordinate was defined as , $\frac{d_v}{d_v+d_d}$ ,where $d_v$ and $d_d$ represent the distance from the transposed injection to the ventral and dorsal border respectively. The rostrocaudal coordinate was obtained by first calculating a line along the rostrocaudal extend of A29 and A30, positioned equally distant from their respective dorsal and ventral borders. Next, we calculated the shortest vector between the transposed injection and the line and found the intersection between the two. The rostrocaudal coordinate was defined as $\frac{d_r}{d_r+d_c}$, where $d_r$ and $d_c$ is the cumulative distance from the cross section to the rostral and caudal end of RSC. This resulted in the normalized map of the positions of all injection sites analyzed in this study (*Figure 1D*).

## Statistical analyses

For cluster analyses, we smoothed each flatmap by applying a Gaussian filter of 5 x 5 bins with a standard deviation of 1.5. Thereafter we calculated pairwise correlations between each possible pair of flatmaps. The correlation matrix was subsequently clustered by using the k-means clustering algorithm in MatLab. The number of clusters was subjectively decided by evaluating the ratio between within- and between cluster variance. The number of clusters was fixed when adding more clusters would not result in a substantial decrease in the within- to between cluster variance ratio.

Our dataset contained experiments in animals of all ages between P1 and P19 and injections in animals aged P27 and P28. Since we did not have samples between P19 and P27 we only included experiments in animals aged P19 or younger in the statistical analyses. Experiments that did not result in labeled axons in any of the areas of PHC where not included in the analyses. The dorsoventral and transverse coordinates of centers of mass were thereafter analyzed independently. For each of the coordinates, we fitted a linear multiple regression model to the data using age and the normalized location of the injection as predictors. A model including the main predictors and two-way interactions was fitted to the data using SPSS (version 20, IBM, Armonk, NY). We subsequently removed from the model the two-way interaction with the lowest standardized β-value and were insignificantly different from zero (t-test), and a new model was fitted. This procedure was repeated until the model only consisted of the main effects and significant two-way interaction terms. In the regression models, the center of mass coordinates ranged from 0 (ventral) to 1 (dorsal) and from 0 (proximal and medial) to 1 (distal and lateral), the coordinates of the injection cite ranged from 0 (rostral) and 1 (caudal) and from 0 (ventral) and 1 (dorsal) and the age of the animal was measured in postnatal age in days. Regression coefficients are reported unstandardized with 95% confidence intervals.

To assess whether the data met the assumption of linear relationships, we visually assessed scatter plots in which each of the independent variables were plotted versus each of the center of mass coordinates. Additionally, we evaluated the standardized predictor values plotted against the

standardized regression residuals to assess whether the assumptions of linearity, homoscedasticity, independent residuals and normally distributed residuals were met. We additionally tested whether the residuals were normally distributed by visually inspecting frequency histograms and normal probability plots. Next, we tested multicollinearity of the independent variables by correlating each of the independent variables. No multicollinearity was assumed if the Pearson's correlations were between -0.70 and 0.70. In addition, we assessed the variance inflation factor (VIF) of each model. The highest calculated VIF was 1.197, which is well below recommended cutoffs. Statistical significance of regression coefficients was determined using two sided $t$-test with $p < 0.05$ as criterion.

## Acknowledgements

We thank Bruno Monterotti and Paulo Girão for technical assistance with experiments. We would also like to thank Kally O'Reilly for sharing the retrograde experiments shown in *Figure 9*. The adult anterograde experiments were performed by Bethany Jones, as part of her PhD-work with MPW who owns all rights to the material.

## Additional information

### Competing interests

MPW: Member of the board of the Kavli Centre, and of the scientific advisory board of the Center for Behavioral Brain Sciences, Otto von Guericke University, Magdeburg, FDR. The other author declares that no competing interests exist.

### Funding

| Funder | Grant reference number | Author |
| --- | --- | --- |
| Norges Forskningsråd | 145993 | Menno P Witter |
| Norges Forskningsråd | 181676 | Menno P Witter |
| Norges Forskningsråd | Centre of Excellence grant | Menno P Witter |
| Kavli Foundation | | Menno P Witter |

The funders had no role in study design, data collection and interpretation, or the decision to submit the work for publication.

### Author contributions

JS, Conception and design, Acquisition of data, Analysis and interpretation of data, Drafting or revising the article; MPW, Conception and design, Analysis and interpretation of data, Drafting or revising the article

### Author ORCIDs

Menno P Witter, http://orcid.org/0000-0003-0285-1637

### Ethics

Animal experimentation: The experimental protocols followed the European Communities Council Directive and the Norwegian Experiments on Animals Act and local directives of the responsible veterinarian at the Norwegian University of Science and Technology. The experimental protocols were approved by the Norwegian Food Safety Authority (#594). All surgeries were conducted under isoflurane gas anesthesia and every effort was made to minimize suffering.

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
