## [Decision Letter]

Thank you for submitting your work entitled "Postnatal development of retrosplenial projections to the parahippocampal region of the rat" for consideration by *eLife*. Your article has been reviewed by three peer reviewers, and the evaluation has been overseen by Howard Eichenbaum as the Reviewing Editor and David Van Essen as the Senior Editor.

The following individuals involved in review of your submission have agreed to reveal their identity: Thomas Van Groen (peer reviewer).

The reviewers have discussed the reviews with one another and the Reviewing Editor has drafted this decision to help you

Summary:

All three reviewers were highly enthusiastic about this study and applauded the meticulous methods and analyses, and felt the results are of high value for understanding the functional circuitry of the cortical-hippocampal system. At the same time, each felt the articulation could be improved for interested readers, and their specific recommendations are listed below.

*Reviewer #1:*

The main issue with the paper is readability, particularly the Methods and Results. The Methods are initially tough to wade through and to understand. I do think the study was well done; the authors successfully met the challenges of comparing anatomical data across developmental ages. However, most readers will not put in the time to access the Methods. Figure 1 is not mentioned until the end of the paper in the Methods. One solution might be to put a paragraph at the beginning of the Results section that explains the methods in conceptual terms, referring to the more detailed text in the Methods. This way the reader can at least understand the subsequent figures in a conceptual way or can jump to the Methods for more detail.

The very detailed results are also difficult to take in. I think readability could be improved by inserting clear summaries are the ends of sections.

There are already a lot of figures, but a schematic summary might also help with readability – this could be a simple schematic or a more complicated wiring diagram. It is also possible that the same end could be achieved with a table.

*Reviewer #2:*

1) The authors discuss at length the involvement and importance of RSC for physiological correlates such as grid cells, head direction cells, etc. However, they only very briefly refer to the body of work that has revealed a causal role for RSC in spatial memory. Moreover, there is no mention of other, emerging evidence of the important for RSC in learning and memory beyond spatial tasks (e.g. work by Corcoran et al., Bucci and colleagues, Helmstetter's group). I suggest there be a least a little more mention of these sorts of studies simply because it would inform the reader as to why the current anatomical study is so important.

2) The authors previously report that there are connections between RSC and perirhinal cortex in their 2011 paper that described the RSC connectome. However, here they report that no such connections were revealed (subsection “General projection patterns”). This apparent conflict should be resolved.

3) As the authors note in the aforementioned 2011 connectome paper, various terminologies have been used to describe/identify the purported subregions of RSC. Here, the authors use the A29/30 nomenclature. It would be useful to briefly mention how those designations relate to other nomenclatures used to distinguish areas within RSC (e.g., by Burwell/Amaral, Vogt).

4) What was the rationale for defining the different age groups here (e.g., in the first paragraph of the subsection “Projection patterns in different age groups”)? It appears to be 1 week, 2 weeks, 3 weeks and older – why is that an appropriate division?

5) It appears that data were pooled from males and females. Any reason to expect (or not) that there are potential sex differences in the temporal maturation of these connections given that there are sex differences in behavior/disorders related to these brain areas?

*Reviewer #3:*

Generally the experiments were performed properly, and are described appropriately. In general, the manuscript is fine, however, quite some improvements can be made to the manuscript. The use of A30 and A29 is correct, but some explanation about other nomenclature for these regions is needed. Furthermore, A29 consists of more than one region, and connections from the sub-areas are not the same. Therefore, injections should be subdivided into these sub-areas. Figure 10 indicates that all projections to PHR arise from layer 5 in RSC, even so it should be mentioned that injections cover what layers in RSC. By the way, Figure 10 is of too low quality (and from what age are the injections?), also the pre/parasubicular injection seems to only retrogradely label A29. While the illustrations of the paper are fine, for the interested reader it would be of use to have some schematic drawing(s) of the described connections, and the timeline of their development. Finally, the schematic maps (e.g., Figure 5 and further) showing the topography are very hard to read, they should be made more accessible to the average reader.

---

## [Author Response]

*Reviewer #1:*

*The main issue with the paper is readability, particularly the Methods and Results. The Methods are initially tough to wade through and to understand. I do think the study was well done; the authors successfully met the challenges of comparing anatomical data across developmental ages. However, most readers will not put in the time to access the Methods. Figure 1 is not mentioned until the end of the paper in the Methods. One solution might be to put a paragraph at the beginning of the Results section that explains the methods in conceptual terms, referring to the more detailed text in the Methods. This way the reader can at least understand the subsequent figures in a conceptual way or can jump to the Methods for more detail.*We appreciate the positive comments. We agree that the Methods are complicated. We have inserted short “conceptual” paragraphs before the description of the normalization of injections (subsection “Injection sites”, last paragraph) and before descriptions of the flatmaps (subsection “Specific projection patterns from different parts of RSC”, first paragraph). We have also reorganized the sequence of figures such that the original Figure 1 is now referred to in the Results in an appropriate place as Figure 4.

*The very detailed results are also difficult to take in. I think readability could be improved by inserting clear summaries are the ends of sections.*

We appreciate this constructive suggestion. Summary statements are now inserted at the end of all relevant paragraphs, aiming to also compare the outcome of a particular analysis to that described previously.

*There are already a lot of figures, but a schematic summary might also help with readability –*

*this could be a simple schematic or a more complicated wiring diagram. It is also possible that the same end could be achieved with a table.* We thank the reviewer for this excellent suggestion. We have prepared a new Figure 10, which schematically summarizes the main findings of the manuscript.

*Reviewer #2:*

*1) The authors discuss at length the involvement and importance of RSC for physiological correlates such as grid cells, head direction cells, etc. However, they only very briefly refer to the body of work that has revealed a causal role for RSC in spatial memory. Moreover, there is no mention of other, emerging evidence of the important for RSC in learning and memory beyond spatial tasks (e.g. work by Corcoran et al., Bucci and colleagues, Helmstetter's group). I suggest there be a least a little more mention of these sorts of studies simply because it would inform the reader as to why the current anatomical study is so important.* We agree that the functional relevance of RSC in memory processes was under represented in the manuscript. We added a few sentences (Introduction, last paragraph) to emphasize these features of RSC. We decided to not bring this point into the Discussion since there are currently few studies investigating the functional relevance of the different RSC subdivisions. In addition, there are no developmental data exploring this topic. It is therefore challenging to discuss our results in this aspect.

*2) The authors previously report that there are connections between RSC and perirhinal cortex in their 2011 paper that described the RSC connectome. However, here they report that no such connections were revealed (subsection “General projection patterns”). This apparent conflict should be resolved.*

This connection has previously been reported in Burwell and Amaral (Cortical afferents of the perirhinal, postrhinal, and entorhinal cortices of the rat. J Comp Neurol. 1998 Aug 24;398(2):179-205). The authors observed extremely few retrogradely labeled cell bodies in RSC after injections in A35 and 36. We have clearly stated the controversy in the subsection “General projection patterns”. Since this is only a small detail, we have dealt with this when describing the lack of projections in PER. In the Burwell paper, retrograde tracers were used. In our hands, retrograde tracers are generally more sensitive, but less specific than iontophoretically injected anterograde tracers, therefore the labeling in perirhinal cortex might have been the result of transport by passing fibers or we may have missed very sparse and relatively long projections by using anterograde tracers and relatively short post-injection survival times.

*3) As the authors note in the aforementioned 2011 connectome paper, various terminologies have been used to describe/identify the purported subregions of RSC. Here, the authors use the A29/30 nomenclature. It would be useful to briefly mention how those designations relate to other nomenclatures used to distinguish areas within RSC (e.g., by Burwell/Amaral, Vogt).* We inserted a new paragraph in the beginning of the Results. We refer to Vogt for the nomenclature and to Sugar 2011 for a rosetta stone for RSC nomenclatures.

*4) What was the rationale for defining the different age groups here (e.g., in the first paragraph of the subsection “Projection patterns in different age groups”)? It appears to be 1 week, 2 weeks, 3 weeks and older – why is that an appropriate division?* These groups were only assigned for visualization purposes. The statistical analysis was done using a continuous age scale.

We chose to visualize these periods because the intrinsic HF-PHR connectivity has been reported to be fully developed during the second postnatal week (before P13), while the functional connectivity is fully developed from P14 an onwards. In addition, eye-opening occurs around P14 which is presumed to be an important developmental milestone. We therefore felt it was natural to group animals into P1-P6, P7-P13 and P14+.

*5) It appears that data were pooled from males and females. Any reason to expect (or not) that there are potential sex differences in the temporal maturation of these connections given that there are sex differences in behavior/disorders related to these brain areas?* This was done to save animals. In each litter we obtained approximately 5 females and 5 males. To avoid having five unused animals from each litter we chose to do experiments on both genders. We thereby reduced the numbers of litters by 50%. In the literature, there are no strong data indicative of sex differences, neither in development nor in connectivity of the areas of research.

*Reviewer #3: Generally the experiments were performed properly, and are described appropriately. In general, the manuscript is fine, however, quite some improvements can be made to the manuscript. The use of A30 and A29 is correct, but some explanation about other nomenclature for these regions is needed.*

Thanks for this overall very positive evaluation. As stated above, we inserted a new paragraph in the beginning of the Results where the nomenclature and the cytoarchitectonic criteria used to define these areas is described.

*Furthermore, A29 consists of more than one region, and connections from the sub-areas are not the same. Therefore, injections should be subdivided into these sub-areas.*

We totally agree with this comment. However, to subdivide A29 further is impossible in the youngest pups since the cytoarchitectonic differences are not apparent at this age. We therefore chose to define the dorsoventral axis of RSC as a continuous measure. The reasoning for this is that a continuous “subdivision” is comparable across all ages while the classical cytoarchitectonic subdivision is only usable for the oldest pups. We have now inserted a statement on this both in the Results section (first paragraph) and in the Methods section (subsection “Location of labeled fibers”, end of second paragraph). We acknowledge that our continuous measure is only indirectly related to the classical cytoarchitectonic subdivision since the classical borders of A29a, b and c follow only approximately our continuous definition of the dorsoventral axis of RSC. In our material we find that the transverse coordinate of center of mass of the RSC projections to PHR show an approximately linear relationship with the dorsoventral coordinate of the injection in RSC. We did not observe any discrete “jumps”. This suggest that the topographical organization of projections from RSC to PHR is not organized into discrete projection patterns from each of A29a, b or c (or A30) but is rather organized in a continuous gradient over the dorsoventral RSC, similar to what has been reported for the adult situation.

Figure 10 indicates that all projections to PHR arise from layer 5 in RSC, even so it should be mentioned that injections cover what layers in RSC.

This is an important point, which was stated in the original manuscript and we also now explicitly mention this in the subsection “Development of density of projection”.

By the way, Figure 10 is of too low quality (and from what age are the injections?),

We expect that this comment resulted from the fairly low resolution of the figure we initially embedded in the manuscript. We have now uploaded a new version of this figure with increased resolution, such that all figures are now at the same, high resolution. The ages of the animals are described in the aforementioned subsection and in the legend to Figure 8. We have also added the ages into the figure.

*also the pre/parasubicular injection seems to only retrogradely label A29.*

The injection in this animal is relatively large compared to the size of the brain and covers PrS, PaS and MEC. There are labeled neurons in both A30 and A29; however, it is clear that the neurons with the strongest fluorescent signal are located ventral A29.

*While the illustrations of the paper are fine, for the interested reader it would be of use to have some schematic drawing(s) of the described connections, and the timeline of their development.*

This is a very good suggestion. We have submitted a Figure 10, which schematically summarize the main findings of the manuscript.

Finally, the schematic maps (e.g., Figure 5 and further) showing the topography are very hard to read, they should be made more accessible to the average reader.

We agree to this comment. We have therefore reorganized the figures, moving the former Figure 5, Figure 6 and Figure 7 into supplementary figures. We have prepared new Figure 5 and Figure 6, mapping the centers of mass of labelling for all experiments, which hopefully are more simple to read.